# SoftCVI: Contrastive variational inference with self-generated soft labels

**Daniel Ward**[1]**, Mark Beaumont**[2]**, Matteo Fasiolo**[1]
[1]School of Mathematics, Bristol University, UK
[2]School of Biological Sciences, Bristol University, UK

## Abstract

Estimating a distribution given access to its unnormalized density is pivotal in Bayesian inference, where the posterior is generally known only up to an unknown normalizing constant. Variational inference and Markov chain Monte Carlo methods are the predominant tools for this task; however, both are often challenging to apply reliably, particularly when the posterior has complex geometry. Here, we introduce Soft Contrastive Variational Inference (SoftCVI), which allows a family of variational objectives to be derived through a contrastive estimation framework. The approach parameterizes a classifier in terms of a variational distribution, re-framing the inference task as a contrastive estimation problem aiming to identify a single true posterior sample among a set of samples. Despite this framing, we do not require positive or negative samples, but rather learn by sampling the variational distribution and computing ground truth soft classification labels from the unnormalized posterior itself. The objectives have zero variance gradient when the variational approximation is exact, without the need for specialized gradient estimators. We empirically investigate the performance on a variety of Bayesian inference tasks, using both simple (e.g. normal) and expressive (normalizing flow) variational distributions. We find that SoftCVI can be used to form objectives which are stable to train and mass-covering, frequently outperforming inference with other variational approaches.

## 1 Introduction

Consider a probabilistic model with a set of parameters $\boldsymbol{\theta}$, for which given a set of observations $\boldsymbol{x}_{\text{obs}}$ we wish to infer a posterior $p(\boldsymbol{\theta}|\boldsymbol{x}_{\text{obs}})$. Unless the model takes a particularly convenient form, the posterior cannot be directly computed. However, typically $p(\boldsymbol{\theta}, \boldsymbol{x}_{\text{obs}})$ is available, related to the posterior by $p(\boldsymbol{\theta}|\boldsymbol{x}_{\text{obs}}) = p(\boldsymbol{\theta}, \boldsymbol{x}_{\text{obs}})/p(\boldsymbol{x}_{\text{obs}})$, where $p(\boldsymbol{x}_{\text{obs}})$ is an intractable normalizing constant $p(\boldsymbol{x}_{\text{obs}}) = \int p(\boldsymbol{\theta}, \boldsymbol{x}_{\text{obs}})d\boldsymbol{\theta}$. In these cases, computational methods such as Markov chain Monte Carlo (MCMC) (Hastings, 1970; Metropolis et al., 1953) or variational inference (Jordan et al., 1999; Kingma & Welling, 2014) are required to perform inference.

Variational inference approaches the inference task as an optimization problem by defining a variational distribution $q_{\boldsymbol{\phi}}(\boldsymbol{\theta})$ and minimizing its divergence to the true posterior $p(\boldsymbol{\theta}|\boldsymbol{x}_{\text{obs}})$. The performance and reliability of variational inference is dependent on numerous factors, including the choice of divergence, variational distribution and parameter initialization. Whilst choosing a divergence that favors mass-covering posterior estimates may facilitate reliable inference, in many cases these divergences introduce bias or are less stable to train, leading to worse performance (Dhaka et al., 2021; Naesseth et al., 2020). Alongside difficulties in assessing performance (Yao et al., 2018), these issues hinder practical applications of variational inference, particularly in applications such as scientific research, where accurate uncertainty quantification is crucial.

In contrast to variational inference, contrastive learning is generally used to perform inference when the likelihood function is only available through an intractable integral, such as for fitting energy-based models or for performing simulation-based inference (Gutmann et al., 2022). Learning is achieved by contrasting positive samples with negative samples, with the latter often generated by augmenting positive samples or by drawing samples from a carefully chosen noise distribution. A

common theme is to leverage the invariance of classification-based objectives to unknown normalizing constants to enable learning where likelihood-based methods are infeasible.

**Contribution**

We show that contrastive estimation can be used to derive a family of variational objectives, terming the approach Soft Contrastive Variational Inference (SoftCVI). The task of fitting the posterior approximation is reframed as a classification task aiming to identify a single true posterior sample among a set of samples. Instead of using explicitly positive and negative samples, we show that for arbitrary samples from a proposal distribution, we can generate ground truth soft classification labels using the unnormalized posterior density itself. The samples and corresponding labels are used for fitting a classifier parameterized in terms of the variational distribution, such that the optimal classifier recovers the true posterior. SoftCVI enables derivation of stable, mass-covering objectives, and performance is demonstrated across a series of experiments using both simple and flexible (normalizing flow) variational distributions. Compared to alternative variational objectives, we find SoftCVI generally yields better calibrated posteriors with a lower forward Kullback-Leibler (KL) divergence to the true posterior. We provide a pair of Python packages, pyrox and softcvi_validation, which provide the implementation, and the code for reproducing the results of this paper, respectively.

## 2 SOFTCVI

In order to fit a variational distribution with SoftCVI, we must define a proposal distribution $\pi(\boldsymbol{\theta})$, a negative distribution $p^-(\boldsymbol{\theta})$, and the variational distribution itself, $q_\phi(\boldsymbol{\theta})$. At each optimization step, three steps are performed which allow fitting the variational distribution:

1. Sample parameters $\{\boldsymbol{\theta}_k\}_{k=1}^K \sim \pi(\boldsymbol{\theta})$ from the proposal distribution $\pi(\boldsymbol{\theta})$.

2. Generate corresponding ground truth soft labels $\boldsymbol{y} \in (0,1)^K$ for the task of classifying between positive and negative samples, presumed to be from $p(\boldsymbol{\theta}|\boldsymbol{x}_{\text{obs}})$ and $p^-(\boldsymbol{\theta})$, respectively.

3. Use $\{\boldsymbol{\theta}_k\}_{k=1}^K$ along with the soft labels $\boldsymbol{y}$ to optimize a classifier parameterized in terms of the variational distribution $q_\phi(\boldsymbol{\theta})$, such that the optimal classifier recovers the true posterior.

The choice of proposal distribution in step one will influence the region in which learning is focused. Throughout the experiments here, we take the intuitive and convenient choice of using the variational distribution itself as the proposal distribution $\pi(\boldsymbol{\theta}) = q_\phi(\boldsymbol{\theta})$, which over the course of training directs learning towards regions with reasonable posterior mass. The remainder of this section is structured as follows: section 2.1 details the assignment of ground truth labels to arbitrary proposal samples; section 2.2 discusses the parameterization and optimization of the classifier; finally, section 2.3 considers the choice of negative distribution.

### 2.1 GENERATING GROUND TRUTH SOFT LABELS

Classification can be used to estimate the density ratio between positive and negative distributions (Gutmann & Hyvärinen, 2012; Hastie, 2009; Oord et al., 2018; Thomas et al., 2022). Conversely, if the positive and negative densities are known, the density ratio and ground truth classification labels can be directly computed. Consider we have a set of $\{\boldsymbol{\theta}_k\}_{k=1}^K$ samples from the proposal distribution $\pi(\boldsymbol{\theta})$. SoftCVI reframes inference as a classification task, where the problem is chosen such that analytical ground truth labels can be assigned to the $K$ samples. Specifically, if we consider the true posterior $p(\boldsymbol{\theta}|\boldsymbol{x}_{\text{obs}})$ to be the positive distribution, and $p^-(\boldsymbol{\theta})$ to be a chosen negative distribution, the ratio $p(\boldsymbol{\theta}_k|\boldsymbol{x}_{\text{obs}})/p^-(\boldsymbol{\theta}_k)$ represents the relative likelihood of a sample being a true posterior sample under the classification problem. By contrasting the $K$ samples, assuming a setting where $\{\boldsymbol{\theta}_k\}_{k=1}^K$ consists of exactly one positive sample from the true posterior $p(\boldsymbol{\theta}|\boldsymbol{x}_{\text{obs}})$, and $K-1$ negative samples from $p^-(\boldsymbol{\theta})$, the optimal classifier is then given by

$$y_k = \frac{p(\boldsymbol{\theta}_k|\boldsymbol{x}_{\text{obs}})/p^-(\boldsymbol{\theta}_k)}{\sum_{k'=1}^K p(\boldsymbol{\theta}_{k'}|\boldsymbol{x}_{\text{obs}})/p^-(\boldsymbol{\theta}_{k'})}, \tag{1}$$

where $y_k$ is the probability on the interval $(0,1)$ that $\boldsymbol{\theta}_k$ is the positive sample among all the considered samples, and $\sum_{k=1}^K y_k = 1$. This approach of contrasting samples is invariant to multiplicative

scaling of the density ratio $p(\boldsymbol{\theta}|\boldsymbol{x}_{\text{obs}})/p^-(\boldsymbol{\theta})$, meaning that access only to unnormalized forms of $p(\boldsymbol{\theta}|\boldsymbol{x}_{\text{obs}})$ and $p^-(\boldsymbol{\theta})$ is sufficient for computing the labels. Since typically $p(\boldsymbol{\theta}, \boldsymbol{x}_{\text{obs}})$ is known and proportional to $p(\boldsymbol{\theta}|\boldsymbol{x}_{\text{obs}})$, we can analytically compute the labels as

$$y_k = \frac{p(\boldsymbol{\theta}_k, \boldsymbol{x}_{\text{obs}})/p^-(\boldsymbol{\theta}_k)}{\sum_{k'=1}^{K} p(\boldsymbol{\theta}_{k'}, \boldsymbol{x}_{\text{obs}})/p^-(\boldsymbol{\theta}_{k'})},$$

$$\boldsymbol{y} = \text{softmax}(\boldsymbol{z}), \quad \text{where } z_k = \log \frac{p(\boldsymbol{\theta}_k, \boldsymbol{x}_{\text{obs}})}{p^-(\boldsymbol{\theta}_k)}, \tag{2}$$

where $p^-(\boldsymbol{\theta})$ may be unnormalized. While contrastive estimation methods generally use explicitly positive and negative examples with hard labels $\boldsymbol{y} \in \{0, 1\}^K$, here, in our framework, we lack access to positive samples from $p(\boldsymbol{\theta}|\boldsymbol{x}_{\text{obs}})$, and we may not have access to samples from $p^-(\boldsymbol{\theta})$. Nonetheless, by instead generating samples from a proposal distribution, $\{\boldsymbol{\theta}_k\}_{k=1}^{K} \sim \pi(\boldsymbol{\theta})$, and by assigning soft labels to these samples using eq. (2), it is still possible to train a classifier which at optimality recovers the true posterior, as we will discuss in the subsequent section.

## 2.2 PARAMETERIZATION AND OPTIMIZATION OF THE CLASSIFIER

Analogously to the computation of the labels in eq. (2), we can parameterize the classifier in terms of the ratio between the variational and negative distribution

$$\hat{y}_k = \frac{q_{\boldsymbol{\phi}}(\boldsymbol{\theta}_k)/p^-(\boldsymbol{\theta}_k)}{\sum_{k'=1}^{K} q_{\boldsymbol{\phi}}(\boldsymbol{\theta}_{k'})/p^-(\boldsymbol{\theta}_{k'})},$$

$$\hat{\boldsymbol{y}} = \text{softmax}(\hat{\boldsymbol{z}}), \quad \text{where } \hat{z}_k = \log \frac{q_{\boldsymbol{\phi}}(\boldsymbol{\theta}_k)}{p^-(\boldsymbol{\theta}_k)}. \tag{3}$$

Given a set of samples $\{\boldsymbol{\theta}_k\}_{k=1}^{K} \sim \pi(\boldsymbol{\theta})$ and corresponding labels $\boldsymbol{y}$ computed using eq. (2), an optimization step with respect to $\boldsymbol{\phi}$ can be taken to minimize the softmax cross-entropy loss function

$$\mathcal{L}_{\text{SoftCVI}}(\boldsymbol{\phi}; \{\boldsymbol{\theta}_k\}_{k=1}^{K}, \boldsymbol{y}) = -\sum_{k=1}^{K} y_k \log(\hat{y}_k) = -\sum_{k=1}^{K} y_k \log\left(\frac{q_{\boldsymbol{\phi}}(\boldsymbol{\theta}_k)/p^-(\boldsymbol{\theta}_k)}{\sum_{k'=1}^{K} q_{\boldsymbol{\phi}}(\boldsymbol{\theta}_{k'})/p^-(\boldsymbol{\theta}_{k'})}\right). \tag{4}$$

This is equivalent to maximizing the categorical log-likelihood, or equivalently, minimizing the forward KL divergence between the true categorical label distribution and the predicted distribution (appendix A.4). While a wide variety of divergence measures, such as $f$-divergences, could also be employed, which we briefly discuss in appendix A.4, we leave a detailed exploration of these for future work.

Let $\Theta$ represent the support of the proposal distribution, with $p(\boldsymbol{\theta}|\boldsymbol{x}_{\text{obs}})$ and $q_{\boldsymbol{\phi}}(\boldsymbol{\theta})$ supported on the same set, and $p^-(\boldsymbol{\theta})$ supported on a superset of $\Theta$. Assume that there exists a $\boldsymbol{\phi}$ such that $p(\boldsymbol{\theta}|\boldsymbol{x}_{\text{obs}}) = q_{\boldsymbol{\phi}}(\boldsymbol{\theta})$. The optimal classifier, i.e. which minimizes eq. (4) for all $\{\boldsymbol{\theta}_k\}_{k=1}^{K} \in \Theta^K$, recovers the true posterior, for any valid choice of $K$.[1] This result follows from the fact the optimal classifier must learn the density ratio between the positive and negative distributions up to a constant across $\Theta$

$$p(\boldsymbol{\theta}|\boldsymbol{x}_{\text{obs}})/p^-(\boldsymbol{\theta}) = c \cdot q_{\boldsymbol{\phi}}(\boldsymbol{\theta})/p^-(\boldsymbol{\theta}),$$
$$p(\boldsymbol{\theta}|\boldsymbol{x}_{\text{obs}}) = c \cdot q_{\boldsymbol{\phi}}(\boldsymbol{\theta}).$$

Due to the shared support, integrating both sides over the proposal support $\Theta$ gives

$$\int_{\Theta} p(\boldsymbol{\theta}|\boldsymbol{x}_{\text{obs}}) = c \cdot \int_{\Theta} q_{\boldsymbol{\phi}}(\boldsymbol{\theta}), \tag{5}$$
$$c = 1.$$

Thus the optimal classifier recovers the true posterior

$$p(\boldsymbol{\theta}|\boldsymbol{x}_{\text{obs}}) = q_{\boldsymbol{\phi}}(\boldsymbol{\theta}). \tag{6}$$

---

[1]Or equivalently, the optimal classifier minimizes $\mathbb{E}_{\{\boldsymbol{\theta}_k\}_{k=1}^{K} \sim \pi(\boldsymbol{\theta})}[\mathcal{L}_{\text{SoftCVI}}(\boldsymbol{\phi}; \{\boldsymbol{\theta}_k\}_{k=1}^{K}, \boldsymbol{y})]$, for which we can view eq. (4) as a single sample Monte Carlo approximation.

Importantly, this suggests if $q_{\phi}(\boldsymbol{\theta})$ or $p(\boldsymbol{\theta}|\boldsymbol{x}_{\text{obs}})$ have non-zero regions outside the support of $\pi(\boldsymbol{\theta})$, then $p(\boldsymbol{\theta}|\boldsymbol{x}_{\text{obs}}) = c \cdot q_{\phi}(\boldsymbol{\theta})$ can be satisfied within $\Theta$ without recovering the true posterior, as the integrals in eq. (5) will not both evaluate to one. This is a consequence of the incentive to learn the density ratio (up to a constant) being localized to the sampled region. Ensuring $\pi(\boldsymbol{\theta})$, $q_{\phi}(\boldsymbol{\theta})$ and $p(\boldsymbol{\theta}|\boldsymbol{x}_{\text{obs}})$ are supported on the same set $\Theta$ can be achieved by defining a variational distribution supported on the same set as the posterior $p(\boldsymbol{\theta}|\boldsymbol{x}_{\text{obs}})$, alongside using the variational distribution as the proposal distribution $\pi(\boldsymbol{\theta}) = q_{\phi}(\boldsymbol{\theta})$.

When the optimal classifier is achieved, we show in appendix A.1 that the gradient is zero for any set $\{\boldsymbol{\theta}_k\}_{k=1}^{K} \in \Theta^{K}$, and consequently has zero variance. This is a generally desirable property that is not present in many variational objectives, although specialized gradient estimators have been developed which in some cases can address this issue (Roeder et al., 2017; Tucker et al., 2018). An algorithm outlining the overall approach of SoftCVI is shown in algorithm 1.

---

**Algorithm 1:** SoftCVI

**Inputs:** $p(\boldsymbol{\theta}, \boldsymbol{x}_{\text{obs}})$, $\pi(\boldsymbol{\theta})$, $p^{-}(\boldsymbol{\theta})$, $q_{\phi_1}(\boldsymbol{\theta})$, number of samples $K \geq 2$, optimization steps $N$, learning rate $\eta$

    **for** $i$ **in** $1 : N$ **do**

1         Sample $\{\boldsymbol{\theta}_k\}_{k=1}^{K} \sim \pi(\boldsymbol{\theta})$

2         Compute soft labels $\boldsymbol{y} = \text{softmax}(\boldsymbol{z})$, where $z_k = \log \frac{p(\boldsymbol{\theta}_k, \boldsymbol{x}_{\text{obs}})}{p^{-}(\boldsymbol{\theta}_k)}$

3         Update $\boldsymbol{\phi}_{i+1} = \boldsymbol{\phi}_i - \eta\nabla_{\phi}\mathcal{L}(\boldsymbol{\phi}_i; \{\boldsymbol{\theta}_k\}_{k=1}^{K}, \boldsymbol{y})$ using the cross-entropy loss, eq. (4)

    **end**

---

## 2.3    CHOICE OF THE NEGATIVE DISTRIBUTION

The choice of negative distribution is a well-known challenge in contrastive learning, with the optimal negative distribution often differing significantly from the positive distribution (Chehab et al., 2022). In contrast to standard applications of contrastive estimation, in SoftCVI, the negative distribution is never sampled, meaning the impact of the choice is limited to its influence on the objective's properties. To better understand this, we can rewrite the softmax cross-entropy objective from eq. (4) by separating out the log term

$$\mathcal{L}_{\text{SoftCVI}}(\boldsymbol{\phi}; \{\boldsymbol{\theta}_k\}_{k=1}^{K}, \boldsymbol{y}) = -\sum_{k=1}^{K} y_k \log \frac{q_{\phi}(\boldsymbol{\theta}_k)}{p^{-}(\boldsymbol{\theta}_k)} + \sum_{k=1}^{K} y_k \log \sum_{k'=1}^{K} \frac{q_{\phi}(\boldsymbol{\theta}_{k'})}{p^{-}(\boldsymbol{\theta}_{k'})},$$

$$= -\sum_{k=1}^{K} y_k \log q_{\phi}(\boldsymbol{\theta}_k) + \log \sum_{k=1}^{K} \frac{q_{\phi}(\boldsymbol{\theta}_k)}{p^{-}(\boldsymbol{\theta}_k)} + \text{const}, \qquad (7)$$

where in the last line we remove the denominator from the first term into a constant as it is independent of $\boldsymbol{\phi}$, and use $\sum_{k=1}^{K} y_k = 1$. The first term encourages placing posterior mass on the samples likely to be from the true posterior, and the second term penalizes the sum of the ratios. In addition to appearing in the second term, the negative distribution choice also influences $y_k$ through eq. (2).

We focus on setting $p^{-}(\boldsymbol{\theta}) = \pi(\boldsymbol{\theta})^{\alpha}$, where $\alpha \in [0, 1]$ is a tempering hyperparameter which interpolates between directly using the proposal distribution as the negative distribution when $\alpha = 1$ and using an improper flat negative distribution when $\alpha = 0$. Lower values of $\alpha$ tend to favor more mass-covering solutions by increasing the relative penalization of samples in higher density regions in $q_{\phi}(\boldsymbol{\theta})$ through the second term of eq. (7). However, a too small choice for $\alpha$ can lead to problematically high variances of the log ratios $z_k = \log[p(\boldsymbol{\theta}_k, \boldsymbol{x}_{\text{obs}})/p^{-}(\boldsymbol{\theta}_k)]$ and $\hat{z}_k = \log[q_{\phi}(\boldsymbol{\theta}_k)/p^{-}(\boldsymbol{\theta}_k)]$. Particularly in high-dimensional problems, this can result in labels and predictions with very few non-zero values, meaning few samples contribute significantly to the loss. In terms of eq. (7), the high variance of $\hat{z}_k$ leads to a failure to sufficiently penalize the ratios in lower density regions of $q_{\phi}(\boldsymbol{\theta})$, as $\log \sum_{k=1}^{K} \exp(\hat{z}_k)$ is dominated by the few largest ratios. In practice, we empirically show this leads to "leakage" of mass into regions of negligible posterior density (fig. 2), and a weak signal-to-noise ratio (SNR) (see appendix A.3).

As described in the introduction to section 2, we choose the proposal distribution to equal the variational distribution, meaning the choice above is equivalent to using $p^{-}(\boldsymbol{\theta}) = q_{\phi}(\boldsymbol{\theta})^{\alpha}$. However,

when computing the objective gradient, we treat $p^-(\boldsymbol{\theta})$ as independent of $\phi$, which practically can be achieved by applying the stop-gradient operator present in automatic differentiation packages.[2] In appendix A.7, we also consider parameterizing the negative distribution using $p(\boldsymbol{\theta}, \boldsymbol{x}_{\text{obs}})^\alpha$; however, this choice introduces the problematic ratio $q_\phi(\boldsymbol{\theta})/p(\boldsymbol{\theta}, \boldsymbol{x}_{\text{obs}})$, which leads to favoring of mode-seeking solutions and poorly calibrated posteriors.

# 3 RELATED WORK

## 3.1 VARIATIONAL INFERENCE

Given access to an unnormalized posterior density, $p(\boldsymbol{\theta}, \boldsymbol{x}_{\text{obs}})$, variational inference allows optimizing a variational distribution $q_\phi(\boldsymbol{\theta})$ to approximate the posterior. For certain model classes there exist closed form solutions which can be exploited during optimization (e.g. Blei et al., 2017; Ghahramani & Beal, 2000; Parisi, 1988). However, the lack of broad applicability of these methods hinders the ability of practitioners to freely alter the model and variational family. As such, we instead focus on methods which place minimal restrictions on the model form and variational family.

### 3.1.1 EVIDENCE LOWER BOUND

The most commonly used variational objective is to minimize the negative Evidence Lower Bound (ELBO) or equivalently, minimizing the reverse KL divergence between the posterior approximation and the true posterior

$$
\begin{aligned}
D_{KL}[q_\phi(\boldsymbol{\theta}) \,\|\, p(\boldsymbol{\theta}|\boldsymbol{x}_{\text{obs}})] &= \mathbb{E}_{q_\phi(\boldsymbol{\theta})}\left[\log q_\phi(\boldsymbol{\theta}) - \log p(\boldsymbol{\theta}|\boldsymbol{x}_{\text{obs}})\right], \\
&= \mathbb{E}_{q_\phi(\boldsymbol{\theta})}[\log q_\phi(\boldsymbol{\theta}) - \log p(\boldsymbol{\theta}, \boldsymbol{x}_{\text{obs}})] + \text{const.}
\end{aligned}
\tag{8}
$$

As there is no general closed-form solution for the divergence, a Monte Carlo approximation is often used (Kingma & Welling, 2014)

$$
\mathcal{L}_{\text{ELBO}}(\phi) = \frac{1}{K}\sum_{k=1}^{K}\left[\log q_\phi(\boldsymbol{\theta}_k) - \log p(\boldsymbol{\theta}_k, \boldsymbol{x}_{\text{obs}})\right], \text{ where } \{\boldsymbol{\theta}_k\}_{k=1}^{K} \sim q_\phi(\boldsymbol{\theta}).
\tag{9}
$$

Although straightforward to apply, the reverse KL divergence heavily punishes placing mass in regions with little mass in the true posterior leading to mode-seeking behavior and lighter tails than $p(\boldsymbol{\theta}|\boldsymbol{x}_{\text{obs}})$. In addition to underestimating uncertainty, the light tails may also lead to the approximation performing poorly in downstream tasks, such as when acting as a proposal distribution in importance sampling (Chatterjee & Diaconis, 2018; Gelman & Meng, 1998; Müller et al., 2019b; Yao et al., 2018) or for reparameterizing MCMC (Hoffman et al., 2019).

### 3.1.2 SELF-NORMALIZED IMPORTANCE SAMPLING FORWARD KL DIVERGENCE

In order to address the limitations of the ELBO, numerous alternative objectives have been proposed that encourage more mass-covering behavior, such as the importance weighted ELBO (Burda et al., 2015), the Rényi divergence, (Li & Turner, 2016), $\chi$-divergence (Dieng et al., 2017), and methods targeting the forward KL divergence (Jerfel et al., 2021; Naesseth et al., 2020). In this section, we will focus on the Self-Normalized Importance Sampling Forward Kullback-Leibler (SNIS-fKL) divergence estimator introduced by Jerfel et al. (2021). Specifically, a standard importance weighted estimate of the forward KL divergence is given by

$$
D_{KL}[p(\boldsymbol{\theta}|\boldsymbol{x}_{\text{obs}}) \,\|\, q_\phi(\boldsymbol{\theta})] = \mathbb{E}_{\pi(\boldsymbol{\theta})}\left[w(\boldsymbol{\theta})\log\frac{p(\boldsymbol{\theta}, \boldsymbol{x}_{\text{obs}})}{q_\phi(\boldsymbol{\theta})}\right] + \text{const},
\tag{10}
$$

where $w(\boldsymbol{\theta}) = p(\boldsymbol{\theta}|\boldsymbol{x}_{\text{obs}})/\pi(\boldsymbol{\theta})$ are the importance weights. Computing a set of self-normalized weights with elements $\tilde{w}(\boldsymbol{\theta})_k = \frac{p(\boldsymbol{\theta}_k, \boldsymbol{x}_{\text{obs}})/\pi(\boldsymbol{\theta}_k)}{\sum_{k'=1}^{K} p(\boldsymbol{\theta}_{k'}, \boldsymbol{x}_{\text{obs}})/\pi(\boldsymbol{\theta}_{k'})}$ and using these alongside a Monte Carlo approximation of eq. (10), yields the objective

$$
\mathcal{L}_{\text{SNIS-fKL}}(\phi; \{\boldsymbol{\theta}_k\}_{k=1}^{K}) = \sum_{k=1}^{K} \tilde{w}(\boldsymbol{\theta})_k \log\frac{p(\boldsymbol{\theta}_k, \boldsymbol{x}_{\text{obs}})}{q_\phi(\boldsymbol{\theta}_k)}.
\tag{11}
$$

---

[2]Without preventing gradient flow through $p^-(\boldsymbol{\theta})$, the choice $p^-(\boldsymbol{\theta}) = \pi(\boldsymbol{\theta}) = q_\phi(\boldsymbol{\theta})$ leads to zero gradients as the parameterization of the classifier becomes $q_\phi(\boldsymbol{\theta})/q_\phi(\boldsymbol{\theta})$.

This approach introduces bias in the approximation of the forward KL vanishing with order $\mathcal{O}(1/K)$ (Agapiou et al., 2017). Generally, the proposal is chosen to equal the variational distribution $\pi(\boldsymbol{\theta}) = q_{\boldsymbol{\phi}}(\boldsymbol{\theta})$, and the proposal parameters are held constant under differentiation.

## 3.2 COMPARISON OF SOFTCVI AND SNIS-FKL

In this section, we demonstrate that in the special case of choosing $p^-(\boldsymbol{\theta}) = \pi(\boldsymbol{\theta}) = q_{\boldsymbol{\phi}}(\boldsymbol{\theta})$, both SoftCVI and SNIS-fKL produce gradients that are equal in expectation, but the SoftCVI objective naturally includes a control variate which ensures the variance of the gradient decreases to zero as the variational distribution approaches the true posterior. We use this result to suggest an alternative, lower-variance gradient estimator for the SNIS-fKL objective, exactly equivalent to optimizing the SoftCVI objective with $p^-(\boldsymbol{\theta}) = \pi(\boldsymbol{\theta}) = q_{\boldsymbol{\phi}}(\boldsymbol{\theta})$.

Noticing that from eq. (11), the numerator term does not depend on $\phi$, we can equivalently write the SNIS-fKL objective as

$$\mathcal{L}_{\text{SNIS-fKL}}(\boldsymbol{\phi}; \{\boldsymbol{\theta}_k\}_{k=1}^K) = -\sum_{k=1}^K \tilde{w}(\boldsymbol{\theta})_k \log q_{\boldsymbol{\phi}}(\boldsymbol{\theta}_k) + \text{const.} \tag{12}$$

In the special case under consideration, the self-normalized weights $\tilde{w}(\boldsymbol{\theta})$ from the SNIS-fKL objective, and the ground truth labels $\boldsymbol{y}$ from SoftCVI are computed identically. This allows rewriting the SoftCVI objective from eq. (7) as the sum of the SNIS-fKL objective and the normalization term

$$\mathcal{L}_{\text{SoftCVI}}(\boldsymbol{\phi}; \{\boldsymbol{\theta}_k\}_{k=1}^K) = \mathcal{L}_{\text{SNIS-fKL}}(\boldsymbol{\phi}; \{\boldsymbol{\theta}_k\}_{k=1}^K) + \log \sum_{k=1}^K \frac{q_{\boldsymbol{\phi}}(\boldsymbol{\theta}_k)}{p^-(\boldsymbol{\theta}_k)} + \text{const}, \tag{13}$$

where we show in appendix A.2 that when $p^-(\boldsymbol{\theta}) = q_{\boldsymbol{\phi}}(\boldsymbol{\theta})$, the normalization term gradient is

$$\nabla_{\boldsymbol{\phi}} \log \sum_{k=1}^K \frac{q_{\boldsymbol{\phi}}(\boldsymbol{\theta}_k)}{p^-(\boldsymbol{\theta}_k)} = \frac{1}{K} \sum_{k=1}^K \nabla_{\boldsymbol{\phi}} \log q_{\boldsymbol{\phi}}(\boldsymbol{\theta}_k). \tag{14}$$

Since $\mathbb{E}_{\pi(\boldsymbol{\theta})}[\nabla_{\boldsymbol{\phi}} \log q_{\boldsymbol{\phi}}(\boldsymbol{\theta})] = \mathbb{E}_{q_{\boldsymbol{\phi}}(\boldsymbol{\theta})}[\nabla_{\boldsymbol{\phi}} \log q_{\boldsymbol{\phi}}(\boldsymbol{\theta})] = \mathbf{0}$, the inclusion or omission of the normalization term does not change the gradient in expectation over sets of $\{\boldsymbol{\theta}_k\}_{k=1}^K$, but it significantly influences the variance. As shown in appendix A.1, the gradient variance for SoftCVI decreases to zero as the variational distribution approaches the posterior. In contrast, this implies the variance of the SNIS-fKL objective approaches the variance of $\frac{1}{K} \sum_{k=1}^K \nabla_{\boldsymbol{\phi}} \log q_{\boldsymbol{\phi}}(\boldsymbol{\theta}_k)$, which is positive, scaling inversely with $K$.

In addition to providing a novel perspective of the SNIS-fKL objective as training an (unnormalized) classifier, this result also implies a straightforward modification to form a lower-variance SNIS-fKL gradient estimator

$$\nabla_{\boldsymbol{\phi}} \mathcal{L}_{\text{LV}}(\boldsymbol{\phi}; \{\boldsymbol{\theta}_k\}_{k=1}^K) = \nabla_{\boldsymbol{\phi}} \mathcal{L}_{\text{SNIS-fKL}}(\boldsymbol{\phi}; \{\boldsymbol{\theta}_k\}_{k=1}^K) + \frac{1}{K} \sum_{k=1}^K \nabla_{\boldsymbol{\phi}} \log q_{\boldsymbol{\phi}}(\boldsymbol{\theta}_k), \tag{15}$$

which is exactly equivalent to optimizing the SoftCVI objective with $p^-(\boldsymbol{\theta}) = \pi(\boldsymbol{\theta}) = q_{\boldsymbol{\phi}}(\boldsymbol{\theta})$. This approach of lowering the variance of a gradient estimator by utilizing $\nabla_{\boldsymbol{\phi}} \log q_{\boldsymbol{\phi}}(\boldsymbol{\theta})$ as a control variate has also been applied to other variational objectives, such as the sticking the landing estimator of the ELBO (Roeder et al., 2017; Tucker et al., 2018). There is no guarantee that the gradient variance will be lower in all instances. However, the variance provably reduces to zero as the variational distribution approaches the true posterior, which allows a reasonable SNR to be maintained even when near convergence, which is likely beneficial for performance and stable convergence. We support this claim with the empirical results in section 5, in addition to investigating the signal-to-noise ratio of the objectives in appendix A.3.

## 3.3 CONTRASTIVE LEARNING

Contrastive learning most commonly allows learning of distributions through the comparison of true samples to a set of negative (noise or augmented) samples (Gutmann & Hyvärinen, 2010). Generally,

and notably in contrast to the current work, applications of contrastive learning focus on problems where the likelihood is unavailable, such as for fitting energy-based models (Gao et al., 2020; Gutmann & Hyvärinen, 2010; Gutmann et al., 2022; Rhodes & Gutmann, 2019; Rhodes et al., 2020) or performing simulation-based inference (Durkan et al., 2020; Greenberg et al., 2019; Gutmann et al., 2022; Hermans et al., 2020; Miller et al., 2022; Thomas et al., 2022). Probably the most widely used objective function in contrastive learning is the InfoNCE loss, proposed by Oord et al. (2018). Given a set of samples $\{\boldsymbol{\theta}_k\}_{k=1}^K$, containing a single true sample with index $k^*$, and $K-1$ negative samples, the loss can be computed as

$$\mathcal{L}_{\text{InfoNCE}}(\boldsymbol{\phi}; \{\boldsymbol{\theta}_k\}_{k=1}^K) = -\log \frac{f_{\boldsymbol{\phi}}(\boldsymbol{\theta}_{k^*})}{\sum_{k=1}^K f_{\boldsymbol{\phi}}(\boldsymbol{\theta}_k)}, \tag{16}$$

where $f_{\boldsymbol{\phi}}(\boldsymbol{\theta})$ approximates the ratio between the positive and negative distributions (up to a constant). This is commonly also presented with an additional sum (or expectation) over different sets of $\{\boldsymbol{\theta}_k\}_{k=1}^K$. The InfoNCE loss can be derived from the softmax cross-entropy loss, eq. (4), by inputting a one-hot encoded vector of labels $\boldsymbol{y}$ with $y_{k^*} = 1$ and all other elements 0. This results in only the $k^*$-th summation term being non-zero, recovering the InfoNCE loss.

In contrastive methods for simulation-based inference, parameters are sampled from a proposal distribution $\pi(\boldsymbol{\theta})$, and used to perform simulations. Learning is achieved through comparing positive parameter-output pairs from $p(\boldsymbol{x}|\boldsymbol{\theta})\pi(\boldsymbol{\theta})$, to negative (mismatched) pairs drawn marginally from $\pi(\boldsymbol{x})\pi(\boldsymbol{\theta})$, where $\pi(\boldsymbol{x}) = \int p(\boldsymbol{x}|\boldsymbol{\theta})\pi(\boldsymbol{\theta})d\boldsymbol{\theta}$. Similar to the current work, a normalized approximation to the posterior is often included in the classifier parameterization. Specifically, the classifier is parameterized using the ratio between the approximate posterior and the prior $q_{\boldsymbol{\phi}}(\boldsymbol{\theta}|\boldsymbol{x})/p(\boldsymbol{\theta})$ and is optimized using the InfoNCE objective (Durkan et al., 2019; Greenberg et al., 2019).

Adversarial methods which are closely related to contrastive learning are often used in conjunction with variational inference (Huszár, 2017; Makhzani et al., 2015; Mescheder et al., 2017). A key idea is to rewrite the reverse KL divergence from eq. (8) to include a density ratio, for example

$$D_{KL}[q_{\boldsymbol{\phi}}(\boldsymbol{\theta}) \,\|\, p(\boldsymbol{\theta}|\boldsymbol{x}_{\text{obs}})] = \mathbb{E}_{q_{\boldsymbol{\phi}}(\boldsymbol{\theta})}\left[\log \frac{q_{\boldsymbol{\phi}}(\boldsymbol{\theta})}{p(\boldsymbol{\theta})} - \log p(\boldsymbol{x}_{\text{obs}}|\boldsymbol{\theta})\right] + \text{const.} \tag{17}$$

The ratio is then substituted with an approximation, $f_{\boldsymbol{\psi}}(\boldsymbol{\theta}) \approx q_{\boldsymbol{\phi}}(\boldsymbol{\theta})/p(\boldsymbol{\theta})$, trained using a separate classification objective. This allows the ELBO to be optimized without computing the log density $\log q_{\boldsymbol{\phi}}(\boldsymbol{\theta})$, enabling the use of expressive implicit models for $q_{\boldsymbol{\phi}}(\boldsymbol{\theta})$. In contrast, SoftCVI directly optimizes the variational distribution with a contrastive objective without a separate classification model, though it does require the variational distribution to be explicitly defined.

Recently, there has been investigation of the use of soft (or ranked) labels in contrastive learning, frequently using cross-entropy-like loss functions. In contrast to the current work, these methods focus on improving performance in the standard context of contrastive learning, where evaluation of the likelihood is infeasible. As such, the soft labels cannot be computed exactly, and instead are generated using other methods, such as label smoothing (Hugger & Uhlmann, 2024) or by utilizing a similarity metric such as cosine similarity between embeddings of positive and negative samples (Feng & Patras, 2022; Hoffmann et al., 2022; Park et al., 2024).

## 4 EXPERIMENTS

We focus on Bayesian inference tasks for which reference posterior samples $\{\boldsymbol{\theta}_i^*\}_{i=1}^{N_{\text{ref}}}$ are available to enable reliable assessment of performance. However, in appendices A.5 and A.6, we also consider application of SoftCVI for training variational autoencoders and Bayesian neural networks, respectively. We use $p^-(\boldsymbol{\theta}) = \pi(\boldsymbol{\theta})^\alpha$ and focus results on two choices, $\alpha = 0.75$ and $\alpha = 1$. Performance is compared to using the ELBO or the SNIS-fKL divergence (Jerfel et al., 2021). For each task and objective, 50 independent runs were performed. Where possible (i.e. an analytical posterior is available), different observations $\boldsymbol{x}_{\text{obs}}$ were generated from the model for each run. For tasks where the reference posterior is created through sampling methods we relied on reference posteriors provided by PosteriorDB (Magnusson et al., 2024) or SBI-Benchmark (Lueckmann et al., 2021) and for each run sampled from the available observations if multiple are present. For all methods, $K = 8$ samples from $q_{\boldsymbol{\phi}}(\boldsymbol{\theta})$ were used during computation of the objectives, and 50,000 optimization steps were performed with the Adam optimizer (Kingma & Ba, 2014).

**Software.** Our implementation and experiments made wide use of the python packages JAX (Bradbury et al., 2018), equinox (Kidger & Garcia, 2021), NumPyro (Phan et al., 2019), FlowJAX (Ward, 2024) and optax (DeepMind et al., 2020).

## 4.1 METRICS

**Coverage probabilities.** Posterior coverage probabilities have been widely used to assess the reliability of posteriors (e.g. Cannon et al., 2022; Cook et al., 2006; Hermans et al., 2021; Prangle et al., 2014; Talts et al., 2018; Ward et al., 2022). Given a nominal frequency $\gamma \in [0, 1]$, the metric assesses the frequency at which true posterior samples fall within the $100\gamma\%$ highest posterior density region (credible region) of the approximate posterior. For a given $\gamma$, if the actual frequency exceeds $\gamma$, the posterior is conservative for that coverage probability; if it is lower, then the posterior is overconfident. A posterior is said to be well-calibrated if the actual frequency matches $\gamma$ for any choice of $\gamma$. A well-calibrated or somewhat conservative posterior is needed for drawing reliable scientific conclusions, and as such is an important property to investigate. We estimate the actual coverage frequency for a posterior estimate as

$$\frac{1}{N_{\text{ref}}} \sum_{i=1}^{N_{\text{ref}}} \mathbb{1} \left\{ \boldsymbol{\theta}_i^* \in \text{HDR}_{q_{\boldsymbol{\phi}}(\boldsymbol{\theta}|\boldsymbol{x}_{\text{obs}})}(\gamma) \right\} \tag{18}$$

where $\mathbb{1}$ is the indicator function, and HDR represents the highest posterior density region, inferred using the density quantile approach of Hyndman (1996).

**Log probability of $\boldsymbol{\theta}^*$.** Looking at the probability of either reference posterior samples or the ground truth parameters in a posterior approximation is a common metric for assessing performance (e.g. Greenberg et al., 2019; Lueckmann et al., 2021; Papamakarios & Murray, 2016). We compute this independently for each posterior approximation as follows

$$\frac{1}{N_{\text{ref}}} \sum_{i=1}^{N_{\text{ref}}} \log q_{\boldsymbol{\phi}}(\boldsymbol{\theta}_i^*). \tag{19}$$

This metric also approximates the negative forward KL divergence between the true and approximate posterior up to a constant, which is a quantity known to control error in importance sampling (Chatterjee & Diaconis, 2018)

$$-D_{\text{KL}}[p(\boldsymbol{\theta}|\boldsymbol{x}_{\text{obs}}) \,\|\, q_{\boldsymbol{\phi}}(\boldsymbol{\theta})] = -\mathbb{E}_{p(\boldsymbol{\theta}|\boldsymbol{x}_{\text{obs}})}[\log p(\boldsymbol{\theta}|\boldsymbol{x}_{\text{obs}}) - \log q_{\boldsymbol{\phi}}(\boldsymbol{\theta})],$$

$$\approx \frac{1}{N_{\text{ref}}} \sum_{i=1}^{N_{\text{ref}}} \log q_{\boldsymbol{\phi}}(\boldsymbol{\theta}_i^*) + \text{const}, \tag{20}$$

**Posterior mean accuracy.** A posterior that is overconfident but with the correct posterior mean would perform poorly based on the aforementioned metrics, but can form good point estimates for the parameters $\boldsymbol{\theta}$. To assess this, we choose to measure the accuracy using the negative $L^2$-norm of the standardized difference in posterior means

$$-\left\| \frac{\text{mean}(\boldsymbol{\theta}^*) - \text{mean}(\boldsymbol{\theta})}{\text{std}(\boldsymbol{\theta}^*)} \right\|_2, \tag{21}$$

where $\text{mean}(\boldsymbol{\theta}^*)$ and $\text{mean}(\boldsymbol{\theta})$ are the mean vectors of the reference and posterior approximation samples respectively, and $\text{std}(\boldsymbol{\theta}^*)$ is the vector of standard deviations of the reference samples.

## 4.2 TASKS

A brief description of each model is given below. For a complete description, see appendix A.8.

**Eight schools.** A classic hierarchical Bayesian inference problem, where the aim is to infer the treatment effects of a coaching program applied to eight schools (Gelman et al., 1995; Rubin, 1981). The parameter set is $\boldsymbol{\theta} = \{\mu, \tau, \boldsymbol{m}\}$, where $\mu$ is the average treatment effect across the schools, $\tau$ is the standard deviation of the treatment effects across the schools and $\boldsymbol{m}$ is the eight-dimensional vector of treatment effects for each school. For the posterior approximation $q_{\boldsymbol{\phi}}(\boldsymbol{\theta})$, we use a normal

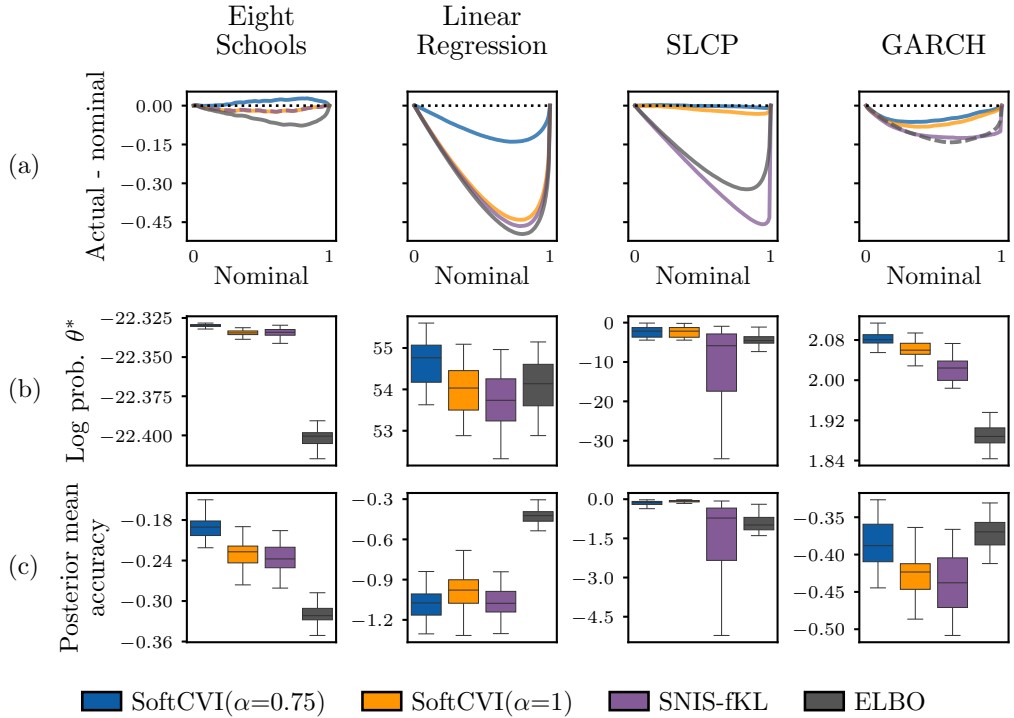

Figure 1: The posterior performance metrics (see section 4.1). a) The nominal coverage frequency against the average difference between the nominal and actual coverage frequency. Well-calibrated methods follow the black dotted line ($y = 0$), whereas conservative methods fall above, and over-confident methods below. b) The average probability of the reference posterior samples in the approximate posterior. c) The accuracy of the approximate posterior mean, calculated as the negative $L^2$-norm between the mean of the standardized reference and approximate posterior samples.

distribution for $\mu$, a folded Student's t distribution for $\tau$ (where folding is equivalent to taking an absolute value transform), and a Student's t distribution for $m$.

**Linear regression.** A Bayesian linear regression model with parameters $\boldsymbol{\theta} = \{\boldsymbol{\beta}, \mu\}$, where $\boldsymbol{\beta} \in \mathbb{R}^{50}$ is the regression coefficients, and $\mu \in \mathbb{R}$ is the bias parameter. The covariates $\boldsymbol{X} \in \mathbb{R}^{(200 \times 50)}$ are sampled from a standard normal distribution, with targets sampled from $\boldsymbol{y} \sim \mathcal{N}(\boldsymbol{X}\boldsymbol{\beta} + \mu, 1)$. The posterior approximation $q_\phi(\boldsymbol{\theta})$ is implemented as a fully factorized normal distribution.

**SLCP.** The Simple Likelihood Complex Posterior (SLCP) task introduced in (Papamakarios et al., 2019). This task parameterizes a multivariate normal distribution using a 5-dimensional vector $\boldsymbol{\theta}$. Due to squaring in the parameterization, the posterior contains four symmetric modes. For the posterior approximation $q_\phi(\boldsymbol{\theta})$, we use a four layer masked autoregressive flow, with a rational quadratic spline transformer (Durkan et al., 2019; Germain et al., 2015; Kingma et al., 2016; Papamakarios et al., 2017).

**GARCH(1,1).** A Generalized Autoregressive Conditional heteroscedasticity (GARCH) model (Bollerslev, 1986). GARCH models are used for modeling the changing volatility of time series data by accounting for time-varying and autocorrelated variance in the error terms. The observation consists of a 200-dimensional time series $\boldsymbol{x}$, where each element $x_t$ is drawn from a normal distribution with mean $\mu$ and time-varying variance $\sigma_t^2$. The variance $\sigma_t^2$ is defined recursively by the update $\sigma_t^2 = \alpha_0 + \alpha_1(x_{t-1} - \mu)^2 + \beta_1 \sigma_{t-1}^2$, where $\alpha_1$ and $\beta_1$ control the contribution from the previous observation and previous variance term, respectively. To parameterize $q_\phi(\boldsymbol{\theta})$, we use a normal distribution for $\mu$ and a log normal distribution for $\alpha_0$. For $\alpha_1$ and $\beta_1$ we use uniform distributions constrained to the prior support, transformed with a rational quadratic spline (Durkan et al., 2019). To allow modeling of posterior dependencies, the distribution over $\beta_1$ is parameterized as a function of $\alpha_0$ and $\alpha_1$ using a neural network.

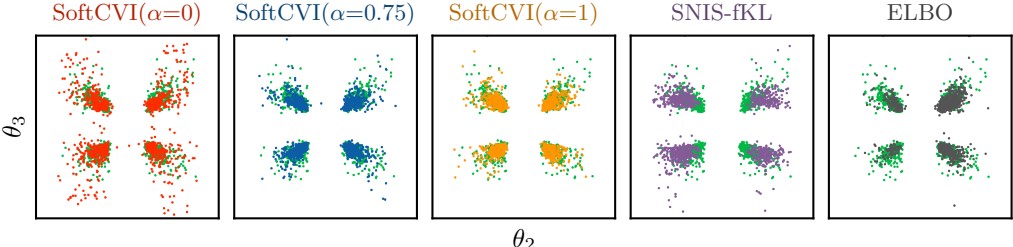

Figure 2: A 2-dimensional posterior marginal for a single run of the SLCP task, with the reference posterior samples shown in green.

## 5 RESULTS

Across all tasks and metrics considered, the SoftCVI derived objectives performed competitively with the ELBO and SNIS-fKL objectives (fig. 1). Overall, SoftCVI using a negative distribution $\pi(\boldsymbol{\theta})^\alpha$ with $\alpha = 0.75$ outperformed the other methods, giving rise to better calibrated posteriors and tending to place more mass on average on the reference posterior samples, indicating a lower forward KL divergence to the true posterior.

A key comparison, is between SoftCVI with $\alpha = 1$ and the SNIS-fKL objective, which give gradient estimators that are equal in expectation (see section 3.2). With the exception of the eight schools task, where both methods performed similarly, SoftCVI with $\alpha = 1$ tended to place more mass on the reference samples and yielded better calibrated posteriors. These results highlight the benefit of the reduced gradient variance provided by SoftCVI when the variational distribution is sufficiently close to the true posterior. Further, the performance discrepancy becomes more pronounced for a smaller choice of $K$ (see figs. 7 and 8 and in the appendix). This can be explained due to the variance of the SNIS-fKL objective gradient scaling inversely with $K$, meaning the control variate naturally included by the SoftCVI objective becomes more crucial. Both SoftCVI and SNIS-fKL tended to place more mass on the reference samples when trained with a larger $K$, but this comes with an increase in computational cost (fig. 7).

On the SLCP task, which yields complex posterior geometry with four symmetric modes, only the SoftCVI objectives performed well. The ELBO objective often resulted in poor distribution of the mass across the modes, sometimes missing modes entirely (fig. 2; see also fig. 9 in the appendix). In contrast, SNIS-fKL, though less mode-seeking, frequently approximated the individual modes poorly. This observation aligns with previous work suggesting the unreliability of existing mass-covering objectives (Dhaka et al., 2021). To illustrate the impact of the choice of negative distribution, fig. 2 also shows a posterior trained with a flat negative distribution by setting $\alpha = 0$. The flat negative distribution does not sufficiently penalize placing mass in $q_\phi(\boldsymbol{\theta})$ in regions of negligible posterior density, leading to "leakage" of mass (see section 2.3).

## 6 CONCLUSION

In this work, we introduced SoftCVI, a novel framework for deriving variational objectives motivated through contrastive learning. Our experiments across various Bayesian inference tasks indicate that SoftCVI often outperforms other variational objectives, producing posterior approximations with better coverage properties and a lower forward KL divergence to the true posterior. The performance difference is particularly notable for tasks with complex posterior geometries which require flexible density estimators. SoftCVI bridges between variational inference and contrastive estimation, which we hope will open up new avenues for further research. Both experimental and theoretical work would be beneficial to further guide the choice of negative distribution. Further, it would be interesting to explore different choices of classification objectives, or equivalently, different choices of divergences between the categorical labels and predictions. Finally, it could be valuable to investigate how advances from classification and contrastive learning, such as label smoothing (Müller et al., 2019a) and temperature scaling (Wang & Liu, 2021), could be adapted to SoftCVI to enhance training stability and further control posterior calibration.

ACKNOWLEDGMENTS

Thank you to Song Liu and Michael Gutmann for providing valuable discussion on this work. Computational facilities were provided by the Advanced Computing Research Centre, University of Bristol - http://www.bristol.ac.uk/acrc/.

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

## A APPENDIX

### A.1 ZERO VARIANCE GRADIENT AT OPTIMUM WITH FINITE K

When the optimal classifier is reached, we can show that the gradient is zero (and hence has zero variance), even for finite $K$. To avoid clashes with the parameters $\phi$, we will use superscripts for indices. We have the objective

$$\mathcal{L}(\phi; \{\boldsymbol{\theta}^k\}_{k=1}^K, \boldsymbol{y}) = -\sum_{k=1}^K y^k \log \left( \frac{q_\phi(\boldsymbol{\theta}^k)/p^-(\boldsymbol{\theta}^k)}{\sum_{k'=1}^K q_\phi(\boldsymbol{\theta}^{k'})/p^-(\boldsymbol{\theta}^{k'})} \right).$$

Letting $\hat{y}_\phi^k$ replace the term inside the log

$$= -\sum_{k=1}^K y^k \log \hat{y}_\phi^k, \tag{22}$$

$$\nabla \mathcal{L}(\phi; \{\boldsymbol{\theta}^k\}_{k=1}^K, \boldsymbol{y}) = -\sum_{k=1}^K y^k \nabla \log \hat{y}_\phi^k, \tag{23}$$

$$= -\sum_{k=1}^K y^k \frac{\nabla \hat{y}_\phi^k}{\hat{y}_\phi^k}, \tag{24}$$

which due to optimality of the classifier we have $y^k = \hat{y}_\phi^k$

$$= -\sum_{k=1}^K \nabla \hat{y}_\phi^k, \tag{25}$$

due to the properties of the softmax, the labels must sum to 1

$$= -\nabla \sum_{k=1}^K \hat{y}_\phi^k = \mathbf{0}. \tag{26}$$

### A.2 SOFTCVI NORMALIZATION TERM GRADIENT

When $p^-(\boldsymbol{\theta}) = q_\phi(\boldsymbol{\theta})$, we claim in eq. (14) that the normalization term gradient can be written as $\frac{1}{K} \sum_{k=1}^K \nabla_\phi \log q_\phi(\boldsymbol{\theta}_k)$. Making use of the property of the logarithmic derivative, $\nabla_{\boldsymbol{x}} \log f(\boldsymbol{x}) = \frac{\nabla_{\boldsymbol{x}} f(\boldsymbol{x})}{f(\boldsymbol{x})}$, we can show this as follows

$$\nabla_\phi \log \sum_{k=1}^K \frac{q_\phi(\boldsymbol{\theta}_k)}{p^-(\boldsymbol{\theta}_k)} = \frac{\nabla_\phi \left[ \sum_{k=1}^K q_\phi(\boldsymbol{\theta}_k)/p^-(\boldsymbol{\theta}_k) \right]}{\sum_{k=1}^K q_\phi(\boldsymbol{\theta}_k)/p^-(\boldsymbol{\theta}_k)},$$

where we treat $p^-(\boldsymbol{\theta}_k)$ as constant with respect to $\phi$ (i.e., we apply stop-gradient), so

$$= \frac{\sum_{k=1}^K \nabla_\phi [q_\phi(\boldsymbol{\theta}_k)]/p^-(\boldsymbol{\theta}_k)}{\sum_{k=1}^K q_\phi(\boldsymbol{\theta}_k)/p^-(\boldsymbol{\theta}_k)},$$

and using $p^-(\boldsymbol{\theta}_k) = q_\phi(\boldsymbol{\theta}_k)$ (value-wise only, not in gradient flow), we get

$$= \frac{\sum_{k=1}^K \nabla_\phi [q_\phi(\boldsymbol{\theta}_k)]/q_\phi(\boldsymbol{\theta}_k)}{\sum_{k=1}^K q_\phi(\boldsymbol{\theta}_k)/q_\phi(\boldsymbol{\theta}_k)},$$

$$= \frac{1}{K} \sum_{k=1}^K \nabla_\phi \log q_\phi(\boldsymbol{\theta}_k). \tag{27}$$

## A.3 GRADIENT SIGNAL-TO-NOISE RATIO

By evaluating an objective gradient over a set of random seeds (different sets of $\{\boldsymbol{\theta}_k\}_{k=1}^K$), it is possible to empirically inspect the signal-to-noise ratio (SNR) ratio of the gradient for different objectives. Following Rainforth et al. (2018), we compute the SNR as

$$\text{SNR}(\nabla_{\boldsymbol{\phi}}\mathcal{L}(\boldsymbol{\phi})) = |\mathbb{E}[\nabla_{\boldsymbol{\phi}}\mathcal{L}(\boldsymbol{\phi})]| \ / \sigma[\nabla_{\boldsymbol{\phi}}\mathcal{L}(\boldsymbol{\phi})]$$

where $\sigma[\cdot]$ denotes the element-wise standard deviation for each parameter gradient. A low SNR indicates that the gradient estimation is dominated by noise, making stochastic optimization challenging. However, it is important to recognize that whilst a high SNR is preferable, it does not alone guarantee the objective itself is practically useful (or even sensible), for example it may be biased or heavily favor mode-seeking solutions.

We consider a toy normal task from Glöckler et al. (2022), with the distinction that we vary the dimensionality of the task $d \in \{1, 50\}$ and the parameterization of variational distribution. The task is defined through the model

$$\boldsymbol{\theta} \sim \mathcal{N}(\mathbf{0}, 4 \cdot \boldsymbol{I}_d), \quad \boldsymbol{x} \sim \mathcal{N}(\boldsymbol{\theta}, \boldsymbol{I}_d),$$

where the aim is to infer the mean vector $\boldsymbol{\theta}$, given an observation $\boldsymbol{x}_{\text{obs}} = \mathbf{1}_d$, where $\mathbf{1}_d$ is the d-dimensional vector of ones. The variational distribution is parameterized as a normal distribution, with mean vector $\boldsymbol{\mu} \in \mathbb{R}^d$ and log standard deviations $\log \boldsymbol{\sigma} \in \mathbb{R}^d$. When assessing the gradient properties of $\boldsymbol{\mu}$, we hold $\log \boldsymbol{\sigma}$ fixed at the closed form posterior solution, $\log \boldsymbol{\sigma} = \log(\sqrt{4/5}) \cdot \mathbf{1}_d$. Similarly, when assessing the gradient properties of $\log \boldsymbol{\sigma}$, we hold $\boldsymbol{\mu}$ fixed at the true closed form solution $\boldsymbol{\mu} = 4/5 \cdot \mathbf{1}_d$. The signal, noise and SNR for each objective are shown in fig. 3.

SoftCVI with $\alpha = 1$, and the SNIS-fKL objective yield very similar signals, which results from the two objectives producing the same gradients in expectation (section 3.2). However, the SNIS-fKL

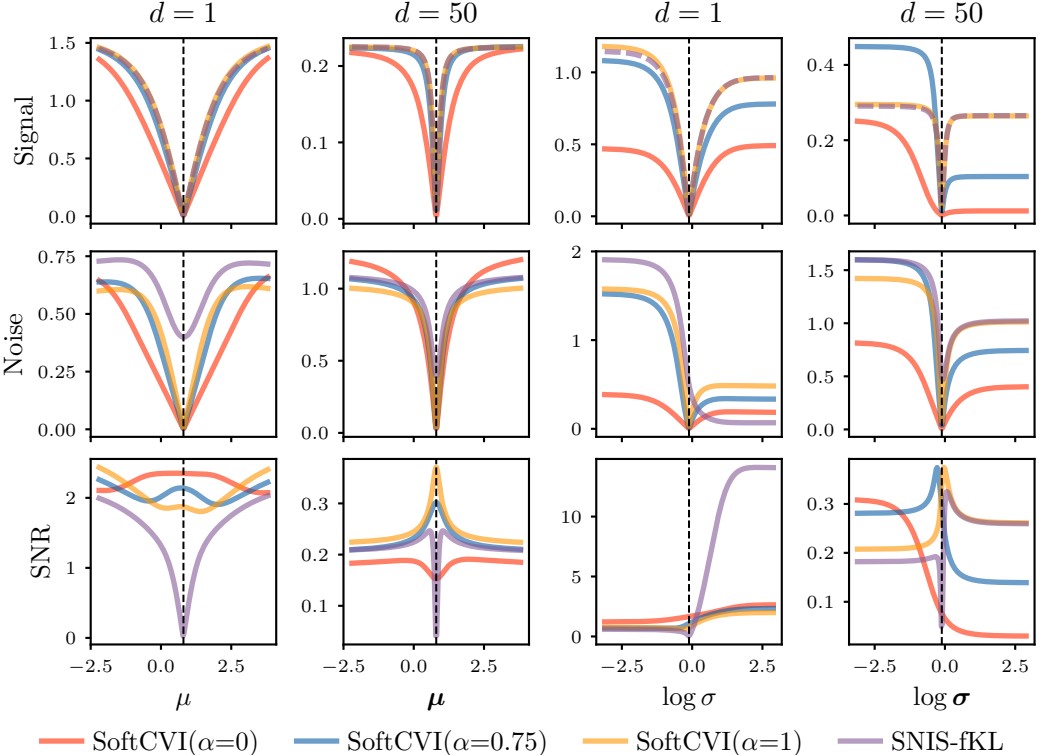

Figure 3: The signal, noise and signal-to-noise ratio of the objective gradients on a toy normal task of varying dimensionality. When $d = 50$, the gradient properties are computed parameter-wise and averaged. The vertical dashed line shows the true parameter values from the closed form posterior solution.

objective has positive noise when the variational distribution approaches the true posterior, meaning the SNR degrades to zero. In contrast, for the SoftCVI objectives, the gradient noise approaches zero (see appendix A.1), and a reasonable SNR is present as the variational distribution approaches the true posterior.

When $d = 50$ the signal and consequently the SNR, deteriorates larger values of $\log \boldsymbol{\sigma}$, and lower choices of $\alpha$. In this case, both expanding the dimension of the problem, and decreasing the $\alpha$ value, increases the variance of $z_k = \log[p(\boldsymbol{\theta}_k, \boldsymbol{x}_{\mathrm{obs}})/p^-(\boldsymbol{\theta}_k)]$ and $\hat{z}_k = \log[q_{\boldsymbol{\phi}}(\boldsymbol{\theta}_k)/p^-(\boldsymbol{\theta}_k)]$. This can lead to degeneracy in the labels and predictions, meaning very few samples meaningfully contribute to the loss function, reducing the SNR. In an extreme case, e.g. setting $\alpha = 0$ and further expanding the dimensionality of the problem, only a single label and prediction will be significantly non-zero. In this regime, negligible signal would exist for a too large $\log \boldsymbol{\sigma}$ parameter, as the variational distribution, despite the incorrect $\log \boldsymbol{\sigma}$, would still result in the correct degenerate predicted labels.

### A.4 ALTERNATIVE DIVERGENCES

We can interpret SoftCVI as defining two categorical distributions between which a divergence is minimized: $P$, which is defined using the labels such that $y_k = P(k)$, and $Q_{\boldsymbol{\phi}}$, which is defined using the predictions such that $\hat{y}_k = Q_{\boldsymbol{\phi}}(k)$. In this work, we used the cross-entropy objective, which is equivalent to minimizing the forward KL-divergence between the label distribution $P$ and the predicted distribution $Q_{\boldsymbol{\phi}}$:

$$
\begin{aligned}
D_{\mathrm{KL}}[P \,\|\, Q_{\boldsymbol{\phi}}] &= \sum_{k=1}^{K} P(k) \log \left( \frac{P(k)}{Q_{\boldsymbol{\phi}}(k)} \right), \\
&= - \sum_{k=1}^{K} P(k) \log \left( Q_{\boldsymbol{\phi}}(k) \right) + \mathrm{const}, \qquad (28) \\
&= - \sum_{k=1}^{K} y_k \log \hat{y}_k + \mathrm{const}.
\end{aligned}
$$

Alternative divergences could be considered for use with SoftCVI. For example, we could explore the general class of $f$-divergences (Ali & Silvey, 1966; Csiszár, 1967), which are defined as:

$$
D_f[P \,\|\, Q_{\boldsymbol{\phi}}] = \sum_{k=1}^{K} Q_{\boldsymbol{\phi}}(k) f \left( \frac{P(k)}{Q_{\boldsymbol{\phi}}(k)} \right),
$$

where $f$ is a convex function $f : (0, \infty) \to \mathbb{R}$, and $f(1) = 0$. Divergences that are more or less mass-covering may be better suited to specific tasks. For example, for many generative modelling tasks reliable uncertainty quantification is considered less critical, in which case a more mode-seeking objective may be favourable. We leave this investigation for future research.

### A.5 TRAINING OF MODEL PARAMETERS: VARIATIONAL AUTOENCODERS

SoftCVI sets up a classification problem which allows fitting the parameters of the variational distribution $\boldsymbol{\phi}$. However, in some contexts, the model may be defined as $p_{\boldsymbol{\psi}}(\boldsymbol{\theta}, \boldsymbol{x}_{\mathrm{obs}})$, with $\boldsymbol{\psi}$, being a set of model parameters which we wish to optimize in some manner alongside $\boldsymbol{\phi}$. SoftCVI does not directly provide a method for fitting such additional model parameters. One approach to resolve this is to include an additional objective, that trains $\boldsymbol{\psi}$ to (approximately) maximize the marginal likelihood.

$$
\mathcal{L}(\boldsymbol{\phi}, \boldsymbol{\psi}; \{\boldsymbol{\theta}_k\}_{k=1}^{K}, \boldsymbol{y}) = \mathcal{L}_{\mathrm{SoftCVI}}(\boldsymbol{\phi}; \{\boldsymbol{\theta}_k\}_{k=1}^{K}, \boldsymbol{y}) + \mathcal{L}_{\mathrm{model}}(\boldsymbol{\psi}; \{\boldsymbol{\theta}_k\}_{k=1}^{K}). \qquad (29)
$$

where in the above formulation we reuse the proposal distribution samples $\{\boldsymbol{\theta}_k\}_{k=1}^{K}$ already sampled for the SoftCVI objective, and we assume the proposal distribution is equal to the variational distribution. One could more generally consider alternating between optimizing the two objectives; however, due to the disjoint parameter sets between the objectives, in addition to the invariance of common optimizers such a Adam to diagonal rescaling of gradient elements (Kingma & Ba, 2014),

we found adding the objectives to work well. In this section, we choose

$$\mathcal{L}_{\text{model}}(\boldsymbol{\psi}; \{\boldsymbol{\theta}_k\}_{k=1}^K) = -\frac{1}{K}\sum_{k=1}^K \log p_{\boldsymbol{\psi}}(\boldsymbol{\theta}_k, \boldsymbol{x}_{\text{obs}}), \tag{30}$$

equivalent to training the model parameters $\boldsymbol{\psi}$ by maximizing the ELBO. As an example, we will train a variational autoencoder (Kingma & Welling, 2014), where $\boldsymbol{\psi}$ includes the parameters of the decoder in addition to the models prior parameters, and $\phi$ consists of the parameters of the encoder. Both the encoder and decoder are parameterized as standard feed-forward networks. The manifolds learned using either the ELBO or the modified SoftCVI objective from eq. (29) are shown in fig. 4, showing qualitatively similar results. Whilst we do not investigate the performance, the SoftCVI method of fitting does not utilize reparameterized gradients and as such is applicable to a broader class of models.

### A.6 BAYESIAN NEURAL ADDITIVE MODEL

Neural additive models enhance the interpretability of neural networks, using an additive model structure (Agarwal et al., 2021). In the simplest case, for a dataset with $C$ features, for an input vector $\boldsymbol{x} = x_1, ..., x_C$, predictions are computed as

$$\hat{y} = \beta + f_1(x_1) + f_2(x_2) + \cdots + f_C(x_C),$$

where each $f_c$ is a neural network corresponding to a single input feature. In this section, we consider a regression problem, in which we parameterize each $f_c$ using a Bayesian neural network (Blundell et al., 2015). Bayesian neural networks have been suggested to reduce the risk of overfitting in addition to providing a method for assessing prediction uncertainty using the inferred posterior predictive distribution. Here, we parameterize each $f_c$ using a single layer Bayesian neural network with a width of either $50$ or $100$, with a Laplace$(0, 1)$ prior on the neural network parameters. This naturally leads to high dimensional posterior distributions ($\boldsymbol{\theta} \in \mathbb{R}^{1500}$ and $\boldsymbol{\theta} \in \mathbb{R}^{3000}$ for the task considered, respectively). We use an independent Gaussian approximation to the posterior.

We consider a regression problem, with synthetic data generated using the nonlinear function

$$y = 0.1 \cdot x_1^3 + |x_2| - x_3 + \epsilon, \quad \text{where } \epsilon \sim \mathcal{N}(0, 3^2)$$

where $\boldsymbol{x} \in \mathbb{R}^{10}$ is generated uniformly from the interval $[-4, 4]$, and the last 7 dimensions are nuisance features. We use 300 training data points, 150 validation data points and 1000 testing data points for computing the metrics.

We assess performance using the average test set log-likelihood under the posterior predictive distribution, and report the mean prediction interquartile range (IQR) across the test set and the associated

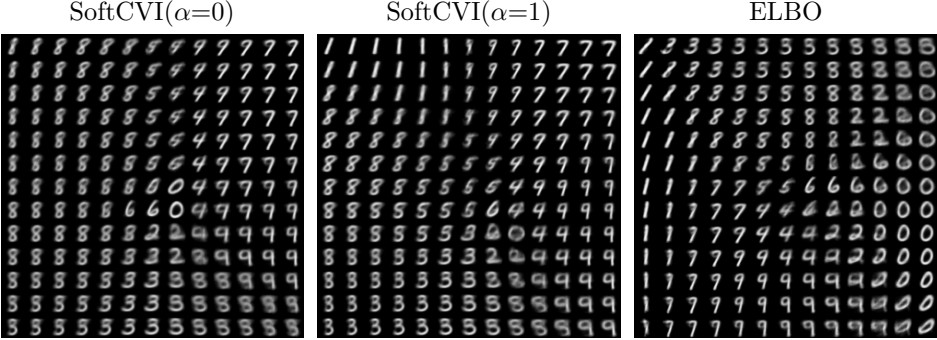

| SoftCVI($\alpha=0$) | SoftCVI($\alpha=1$) | ELBO |
|:---:|:---:|:---:|

Figure 4: The manifolds learned by variational autoencoders on the MNIST dataset, trained using either the ELBO or SoftCVI. To enable training of the model parameters, the SoftCVI objective was modified by adding the model component of the ELBO, $-\frac{1}{K}\sum_{i=1}^K \log p_{\boldsymbol{\psi}}(\boldsymbol{\theta}_k, \boldsymbol{x}_{\text{obs}})$. In all cases, the objectives were trained for 100,000 steps with a batch size of 1, and $K = 8$.

prediction coverage (i.e. the average frequency with which the true underlying function value is included within the predicted IQR). We note there are numerous challenges associated with assessing performance for Bayesian neural networks. For example, high test log-likelihood or calibrated predictions is not necessarily indicative of a good posterior approximation (Yao et al., 2019). Further, we use early stopping based on the validation log-likelihood, in addition to choosing the learning rates with cross-validation, both of which will tend to bias results to favor models with higher test log-likelihood, without consideration of the calibration of the posterior predictive distribution.

We train 10 networks initialized with different random seeds, and report the results in table 1. A plot of the learned components for each method is shown in fig. 5.

| Method | NN Width | Test Log-Likelihood | Prediction IQR | IQR Coverage |
|--------|----------|---------------------|----------------|--------------|
| ELBO | 50 | **-2.392** ± 0.005 | 0.648 ± 0.084 | 0.262 ± 0.042 |
|  | 100 | **-2.392** ± 0.005 | 0.641 ± 0.065 | 0.264 ± 0.031 |
| SNIS-fKL | 50 | -2.422 ± 0.007 | 0.695 ± 0.011 | 0.186 ± 0.008 |
|  | 100 | -2.421 ± 0.005 | 0.938 ± 0.010 | 0.256 ± 0.007 |
| SoftCVI ($\alpha = 0.75$) | 50 | -2.424 ± 0.007 | 0.743 ± 0.018 | 0.202 ± 0.009 |
|  | 100 | -2.423 ± 0.005 | 0.983 ± 0.017 | 0.271 ± 0.010 |
| SoftCVI ($\alpha = 1$) | 50 | -2.422 ± 0.007 | 0.701 ± 0.012 | 0.190 ± 0.009 |
|  | 100 | -2.423 ± 0.005 | 0.943 ± 0.012 | 0.260 ± 0.009 |

Table 1: Performance metrics for the Bayesian neural additive model. Note a model producing calibrated predictions on the test set would yield an IQR coverage value of 0.5.

While all methods demonstrated comparable predictive performance, the ELBO achieved slightly better results, evidenced by the highest test log-likelihood. Both SoftCVI (with $\alpha = 0.75$ or $\alpha = 1$) and SNIS-fKL yielded similar results. Bayesian neural network posteriors are often complex and highly multimodal (Izmailov et al., 2021). In this case, it is likely that SoftCVI does not show a significant advantage over SNIS-fKL because the Gaussian posterior is heavily misspecified. As a result, the approximate posterior may never become sufficiently close to the true posterior in order to provide the variance reduction benefits associated with SoftCVI.

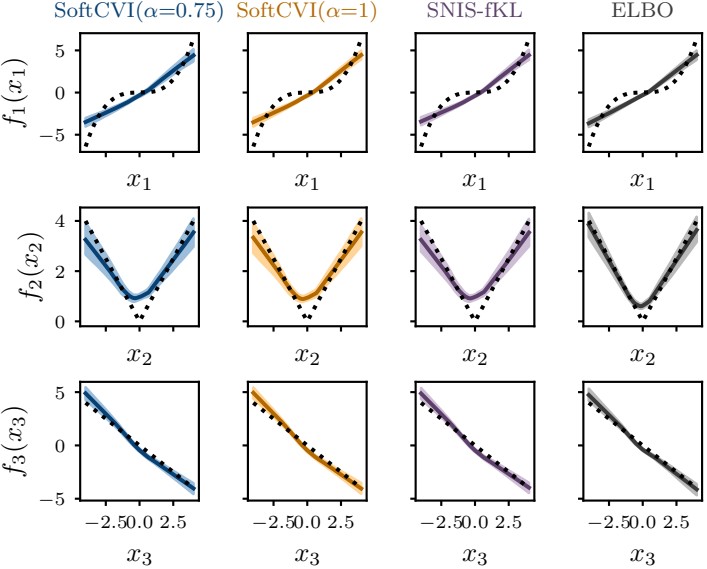

Figure 5: The means and 95% prediction intervals for the components of a Bayesian neural additive model for each method. The true underlying components are shown with the dotted black lines. We restrict to the first three dimensions, ignoring the nuisance variables.

## A.7 ALTERNATIVE NEGATIVE DISTRIBUTION CHOICES

Figure 6: Additional metrics analogous to fig. 1, using $p^-(\boldsymbol{\theta}) = p(\boldsymbol{\theta}, \boldsymbol{x}_{\text{obs}})^{\alpha}$ as the negative distribution choice in the SoftCVI objectives.

The main results focus on parameterizing the negative distribution as a function of the proposal distribution $p^-(\boldsymbol{\theta}) = \pi(\boldsymbol{\theta})^{\alpha}$. However, another possible choice is to use the unnormalized posterior to parameterize the negative distribution, for example $p^-(\boldsymbol{\theta}) = p(\boldsymbol{\theta}, \boldsymbol{x}_{\text{obs}})^{\alpha}$. This choice, when $\alpha = 1$, implies equality between the assumed negative and positive distributions, meaning through eq. (2) the ground truth labels become constant $y_k = \frac{1}{k}$ for $k = 1, ..., K$. Inputting these labels into eq. (7) yields the objective

$$\mathcal{L}(\boldsymbol{\phi}; \{\boldsymbol{\theta}_k\}_{k=1}^K, \boldsymbol{y}) = -\frac{1}{K} \sum_{k=1}^{K} \log q_{\boldsymbol{\phi}}(\boldsymbol{\theta}_k) + \log \sum_{k=1}^{K} \frac{q_{\boldsymbol{\phi}}(\boldsymbol{\theta}_k)}{p^-(\boldsymbol{\theta}_k)} + \text{const.} \qquad (31)$$

Assuming the proposal distribution is chosen to match the variational distribution, since $\mathbb{E}_{\pi(\boldsymbol{\theta})}[\nabla_{\boldsymbol{\phi}} \log q_{\boldsymbol{\phi}}(\boldsymbol{\theta})] = \mathbb{E}_{q_{\boldsymbol{\phi}}(\boldsymbol{\theta})}[\nabla_{\boldsymbol{\phi}} \log q_{\boldsymbol{\phi}}(\boldsymbol{\theta})] = 0$, the first term in eq. (7) has an expected gradient of zero. In contrast to using the proposal distribution as the negative distribution, which results in the normalization term which penalizes the ratios acting as a control variate, here, the first term instead acts as a control variate, with the normalization term providing the signal by penalizing the ratio $q_{\boldsymbol{\phi}}(\boldsymbol{\theta})/p(\boldsymbol{\theta}, \boldsymbol{x}_{\text{obs}}) \propto q_{\boldsymbol{\phi}}(\boldsymbol{\theta})/p(\boldsymbol{\theta}|\boldsymbol{x}_{\text{obs}})$. As such this choice heavily penalizes $q_{\boldsymbol{\phi}}(\boldsymbol{\theta}) \gg p(\boldsymbol{\theta}|\boldsymbol{x}_{\text{obs}})$, favoring overconfident posteriors. We report in fig. 6 the metrics, using $p^-(\boldsymbol{\theta}) = p(\boldsymbol{\theta}, \boldsymbol{x}_{\text{obs}})^{\alpha}$, with $\alpha = 0.75$ and $\alpha = 1$.

## A.8 TASKS

For all tasks, where possible, we make use of reparameterizations to reduce the dependencies between the parameters $\boldsymbol{\theta}$ in the model, and to ensure variables are reasonably scaled, which is generally considered to improve performance (Betancourt & Girolami, 2015; Gorinova et al., 2020; Papaspiliopoulos et al., 2007). Below, we describe the models used and the source of the reference posterior samples.

EIGHT SCHOOLS.

A classic hierarchical inference model (Gelman et al., 1995; Rubin, 1981), aiming to infer the treatment effects of a training program applied to eight schools. The parameter set is $\boldsymbol{\theta} = \{\mu, \tau, \boldsymbol{m}\}$, where $\mu$ is the average treatment effect across the schools, $\tau$ is the standard deviation of the treatment effects across the schools and $\boldsymbol{m}$ is the treatment effects for each school. The model is given by

$$\mu \sim \mathcal{N}(0, 5^2),$$
$$\tau \sim \text{HalfCauchy}(0, 5^2),$$
$$m_i \sim \mathcal{N}(\mu, \tau^2), \quad i = 1, ..., 8,$$
$$x_i \sim \mathcal{N}(m_i, \sigma_i^2), \quad i = 1, ..., 8,$$

where $\boldsymbol{\sigma}^2$ is treated as known, estimated using the standard errors in the data. We use the reference posterior samples available from PosteriorDB, which are sampled using MCMC (Magnusson et al., 2024).

LINEAR REGRESSION.

A linear regression model, defined as

$$\beta_i \sim \mathcal{N}(0, 1), \quad i = 1, ..., 50,$$
$$\mu \sim \mathcal{N}(0, 1),$$
$$x_i \sim \mathcal{N}(\boldsymbol{X}\boldsymbol{\beta} + \mu, 1), \quad j = 1, ..., 200,$$

For each run of the task, we sampled a dataset $\boldsymbol{X} \in \mathbb{R}^{200 \times 50}$ from a standard normal distribution, and drew reference posterior samples from the analytical posterior solution.

SIMPLE LIKELIHOOD COMPLEX POSTERIOR

The SLCP task was introduced by Papamakarios et al. (2019), and is designed to be a challenging inference problem with a multimodal posterior. The data is a set of four samples from a two-dimensional multivariate Gaussian likelihood function. The likelihood is parameterized using $\boldsymbol{\theta}$ using squaring which introduces a complex multimodal structure (see fig. 2). Specifically, the model is defined as

$$\theta_i \sim \text{Uniform}(-3, 3), \quad i = 1, ..., 5 \tag{32}$$
$$\boldsymbol{x}_i \sim \mathcal{N}(\boldsymbol{\theta}_{1:2}, \boldsymbol{\Sigma}), \quad j = 1, ..., 4, \tag{33}$$

where the covariance matrix $\boldsymbol{\Sigma}$ is

$$\boldsymbol{\Sigma} = \begin{bmatrix} s_1^2 & p \cdot s_1 \cdot s_2 \\ p \cdot s_1 \cdot s_2 & s_2^2 \end{bmatrix}, \text{ where } s_1 = \theta_3^2, \ s_2 = \theta_4^2 \text{ and } p = \tanh(\theta_5)$$

Despite being used extensively in the simulation-based inference (SBI) literature, this task has a tractable likelihood so can be used in the current work. The reference posterior is from the SBI Benchmark python package (Lueckmann et al., 2021), and was inferred using sampling/importance resampling using the analytical likelihood function (Rubin, 1988).

GARCH

The GARCH model is a widely used statistical model for analyzing time series data with time-varying volatility (Bollerslev, 1986). It extends the basic autoregressive framework by allowing the conditional variance of the observations to depend on both past observations (controlled via the $\alpha_1$ parameter) and variances (controlled by the $\beta_1$ parameter). GARCH and similar models are often used for modeling financial data, where autocorrelated variances are common. The priors are defined as

$$\mu \sim \text{ImproperUniform}(-\infty, \infty)$$
$$\alpha_0 \sim \text{ImproperUniform}(0, \infty)$$
$$\alpha_1 \sim \text{Uniform}(0, 1)$$
$$\beta_1 \sim \text{Uniform}(0, 1 - \alpha_1),$$

Where ImproperUniform represents an improper flat prior over the specified region. For $t = 1, \ldots, 200$, the variance evolves recursively through the update

$$\sigma_t^2 = \alpha_0 + \alpha_1(x_{t-1} - \mu)^2 + \beta_1 \sigma_{t-1}^2$$

and the likelihood is given by

$$x_t \sim \mathcal{N}(\mu, \sigma_t^2)$$

At initialization, $y_0$ is set to the first observation element, and $\sigma_0^2 = 0.25$. For this task, a reference posterior is available in PosteriorDB, sampled using MCMC (Magnusson et al., 2024).

## A.9 ADDITIONAL FIGURES

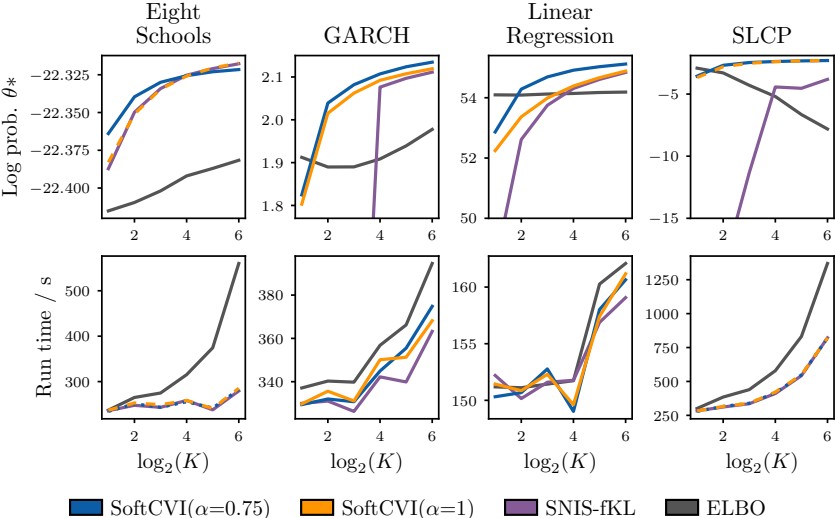

Figure 7: Average log-probability of the reference posterior samples as a function of $K$ (ranging from 2 to 64), along with the associated run times (measured on a CPU with 8GB RAM, including compilation time). The poor performance of SLCP for higher values of $K$ was due to increased mode-seeking behavior. Some results for SNIS-fKL are truncated on the axes to improve visualization of other methods. Note that the run times will be dependent on the architecture of the variational distribution. For example, the ELBO showed significantly slower run times for the SLCP task due to the cost of computing reparameterization gradients for a masked autoregressive flow. However, using an inverse autoregressive flow (Kingma et al., 2016) would likely mitigate this issue.

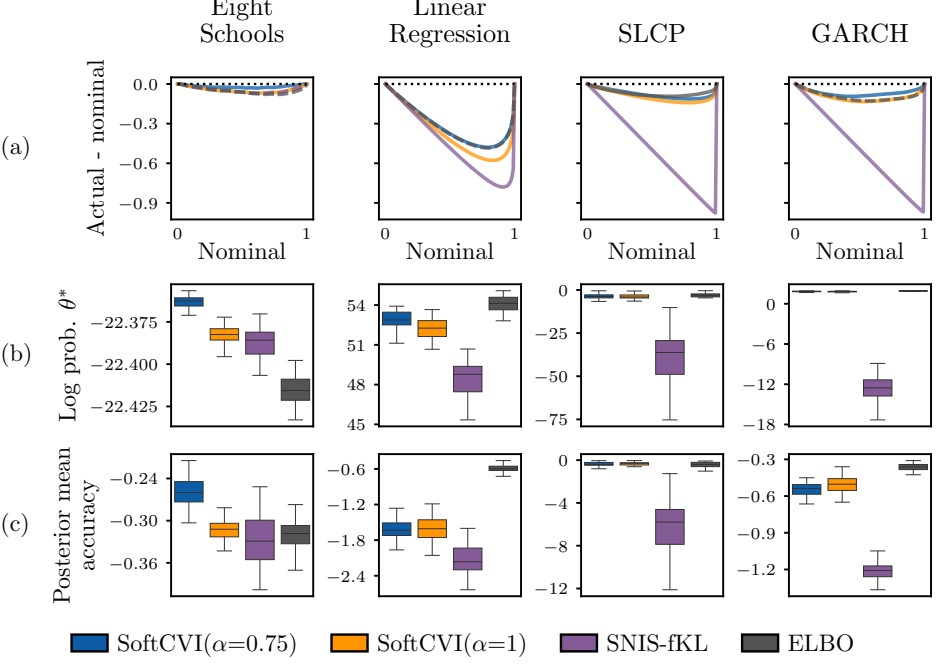

Figure 8: Additional metrics analogous to fig. 1, using $K = 2$, instead of $K = 8$ samples when approximating the objectives. The SNIS-fKL objective performance degrades substantially when $K$ is small.

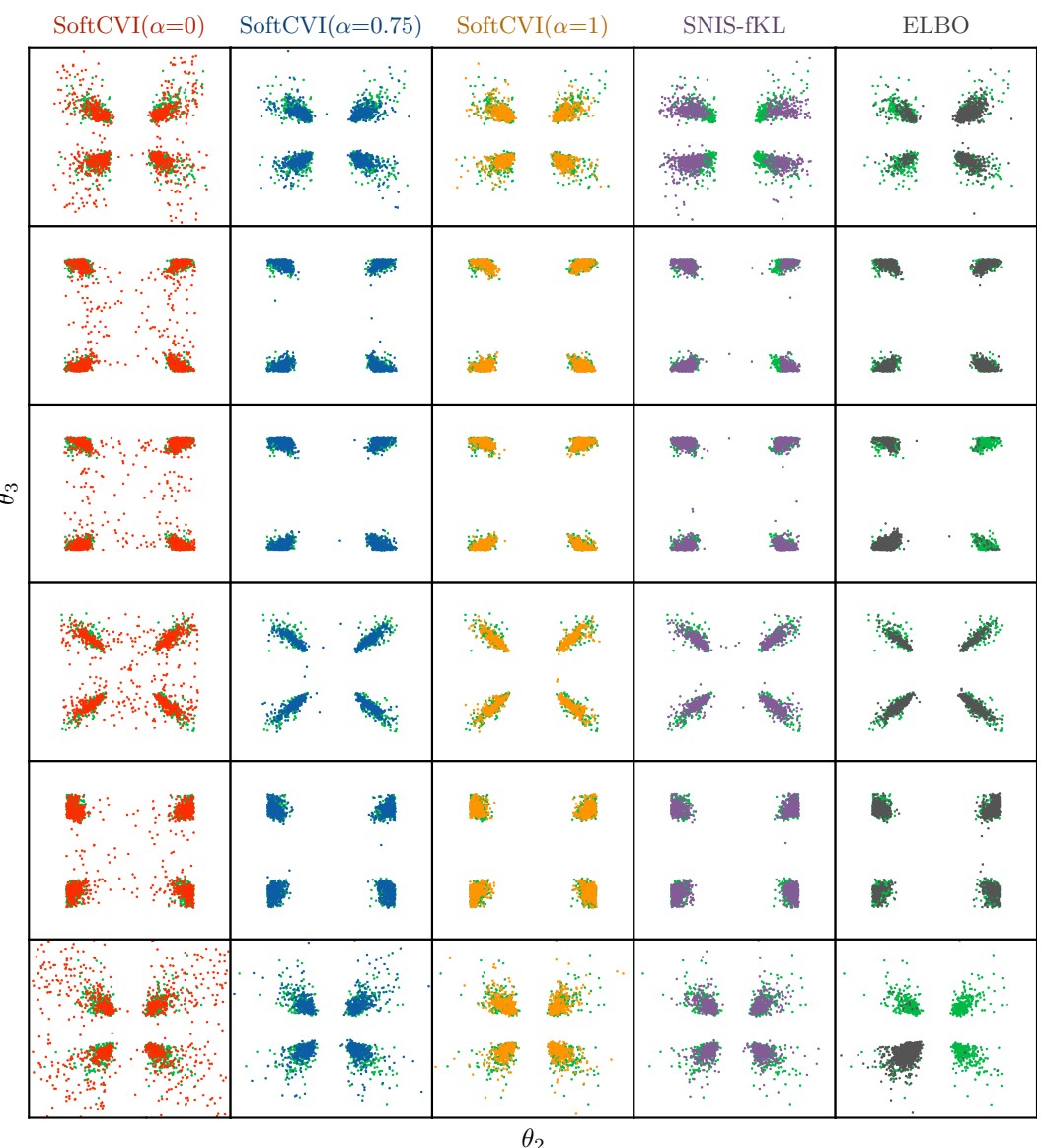

Figure 9: The multimodal 2-dimensional posterior marginals for the first six runs of the SLCP task for each method, with the reference posterior samples shown in green.

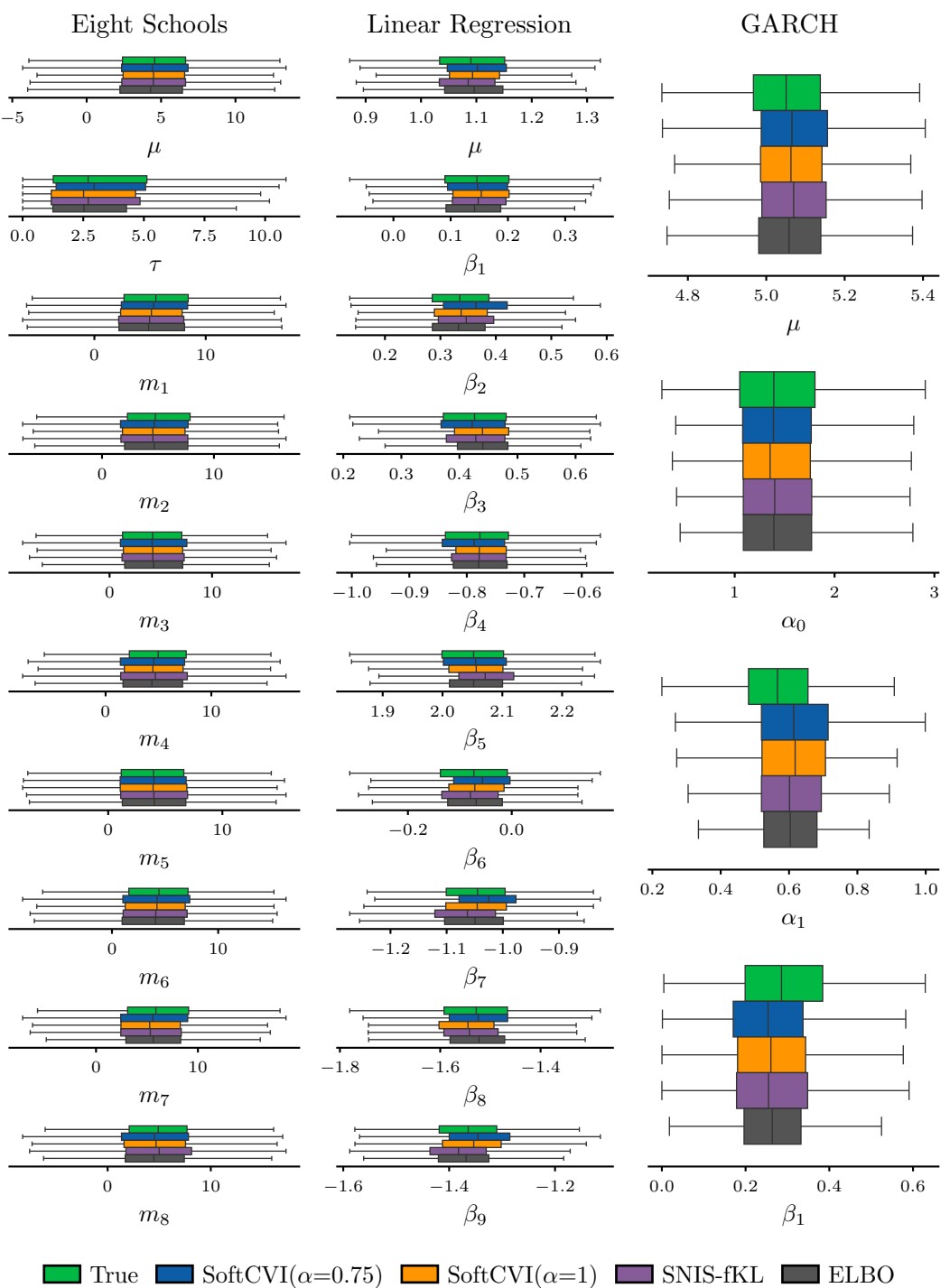

Figure 10: The distribution of the posterior marginals for a single run of the eight schools, linear regression and GARCH(1,1) tasks. For the linear regression task, we restrict the plot to the first 10 parameters of the model.

