# OpenReview forum: "SoftCVI: Contrastive variational inference with self-generated soft labels"
_ICLR.cc/2025/Conference — ICLR 2025 Spotlight_

### Official Review · Reviewer_36B5 · 2024-10-30

**Soundness:** 3
**Presentation:** 4
**Contribution:** 3
**Rating:** 8
**Confidence:** 3

**Summary:**

The paper proposes SoftCVI, a variational inference method based on contrastive learning that uses soft labels, i.e., binary classification probabilities. SoftCVI reframes fitting a variational distribution as finding a classifier to identify actual posterior samples among a set of samples. The paper claims that:
  1. SoftCVI works without having access to samples from the posterior.
  2. SoftCVI forms objectives that are mass-covering.
  3. SoftCVI yields better posterior approximation than baselines with simple and flexible variational distributions.

**Strengths:**

## Reason for score
- The presentation is lucid.
- The derivations look good,  aside from a couple of technicalities (see questions).
- Using soft labels in contrastive learning for Bayesian inference is exciting and novel.
- The connection to SNIS-fKL is clear, and viewing the gradient of eq. 7 as a control variate is insightful and supported by evidence.
- Experimenting with normal and normalizing flow variational distribution demonstrates SoftCVI's flexibility, and the results support that SoftCVI is more mass-covering than ELBO and SNIS-fKL.

**Weaknesses:**

- It is unclear whether the variational distribution is a biased estimator of the posterior. Please see the questions.

**Questions:**

1. Given the connection between SNIS-fKL and SoftCVI and that MC estimation of the SNIS-fKL objective is a biased estimation of the forward KL, does SoftCVI give a biased estimation of the posterior when $p^-(\theta)=\pi(\theta)=q_w(\theta)$? Could you clarify whether SoftCVI with $p^-(\theta)=q(\theta)^\alpha$ gives a biased estimation of the posterior?
2. What is claimed when stating that SoftCVI enables a stable objective (line 064)? Is numerical stability, stable convergence or something else? What evidence is there for the type of stability claimed?
3. Why does the equation after eq.4 hold (line 146)? It seems like there is a population limit ($\lim_{K \rightarrow \infty}$) missing. Is this the case? Otherwise, please explain why this is not necessary.
4. The article establishes that $\alpha=0$ leads to leakage and that $\alpha=0.75$ is an good choice for the experiments. How do you suggest deciding $\alpha$? Could you provide theoretical justification or guidelines for choosing in practice $\alpha$? And, is there a threshold under which $\alpha$ will lead to leakage, or is this only the case for $\alpha=0$?
5. Could you explain the second equality in the derivation leading to eq. 26? For example, by a step-by-step derivation.
6. The claim that the parametric variational distribution can recover the posterior exactly under only a support constraint (lines 143-144) seems odd. Is there a condition that the posterior is contained within the parametric variational family missing? If so, please add this and any other assumption to the paragraph. If not, explain the flaw in the following case: Let the posterior $p(\theta|\mathcal{D})$ be multi-modal (for example, a mixture posterior) and the variational distribution parameterized by $\phi$ be unimodal, e.g., Gaussian, both supported on the real line. By construction, we would have that $\forall \phi.p(\theta|\mathcal{D}) \not = q_\phi(\theta)$ contradicting the claim in lines 143-144 that there $\exists \phi. p(\theta|\mathcal{D}) = q_\phi(\theta)$ namely the $\phi$ that parameterizes the optimal classifier from eq. 3.

---

> ### Author Response · Authors · 2024-11-25
>
> Many thanks for the positive review, and the perceptive questions, which has helped in improving our manuscript.
>
> > Why does the equation after eq. 4 hold (line 146)? It seems like there is a population limit ($\lim_{K\rightarrow\inf}$) missing. Is this the case? Otherwise, please explain why this is not necessary.
>
> Thank you for highlighting the potential ambiguity here. When we discuss the optimal classifier, we mean that the cross entropy objective is minimized for all possible sets $\\{\boldsymbol{\\theta}\_k\\}\_{k=1}^K \in \\Theta^K$
> , in which case $K\geq2$ is sufficient. Alternatively, we can also view the cross-entropy objective used as an unbiased single sample Monte Carlo estimate of
> $$
> E\_{\\{\\boldsymbol{\\theta}\_k\\}\_{k=1}^K \\sim \\pi(\\boldsymbol{\\theta})}
> \\left[
> \\mathcal{L}\_{\\text{SoftCVI}}\\left(\\boldsymbol{\\phi}; \\{\\boldsymbol{\\theta}\_k\\}\_{k=1}^K, \\mathbf{y}\\right)
> \\right]
> $$
> which is minimized when the cross-entropy is minimized for all $\\{\\boldsymbol{\\theta}\_k\\}\_{k=1}^K \\in \\Theta^K$. We have updated the corresponding section to emphasize this information.
>
> > Given the connection between SNIS-fKL and SoftCVI and that MC estimation of the SNIS-fKL objective is a biased estimation of the forward KL, does SoftCVI give a biased estimation of the posterior.
>
> Generally, under the conditions outlined in section 2.2 (including the aforementioned clarification), for any valid choice of $K$ and $p^-(\mathbf{\theta})$, the optimal classifier will recover the true posterior. Selecting $p^-(\mathbf{\theta})$ to be $\pi(\mathbf{\theta})$, if we interpret the gradients from both methods (SNIS-fKL and SoftCVI) as approximations to the gradient of the forward KL, both methods are biased but consistent estimators of the forward KL gradient, with the bias decreasing to zero as the variational distribution approaches the true posterior (or as $K$ tends to infinity). Given the reasonable performance of SoftCVI with $\alpha=1$ and low values of $K$ (see Figs 7-8), we would argue this indicates that SNIS-fKL predominantly performs worse with low values of $K$ due to the increased variance, rather than increased bias.
>
> > The article establishes that $\alpha=0.75$ leads to leakage and that $\alpha=0.75$ is an good choice for the experiments. How do you suggest deciding $\alpha$? Could you provide theoretical justification or guidelines for choosing in practice $\alpha$? And, is there a threshold under which $\alpha$ will lead to leakage, or is this only the case for $\alpha=0$.
>
> In general, the best choice for $\alpha$ will be specific to the task, and will have to be assessed using approaches such as the methods outlined in [1]. However, we hope that by demonstrating competitive performance can be achieved with $\alpha=0.75$ and $\alpha=1$ across all the tasks, including the deep learning examples in appendix A4-5, we have highlighted that precise tuning is not required. Empirically, we did not observe a threshold value of $\alpha$ which leads to leakage, but a smooth transition as $\alpha$ is lowered. We do acknowledge that there is room for further work in theoretically justifying the choice of negative distribution in SoftCVI, and mention this fact in the conclusion.
>
> > What is claimed when stating that SoftCVI enables a stable objective (line 064)? Is numerical stability, stable convergence or something else? What evidence is there for the type of stability claimed?
>
> Primarily stable convergence, which we have evidence for empirically (improved performance, particularly given the link between SoftCVI and SNIS-fKL), and theoretically, by showing the gradient estimator tends towards zero-variance, when the variational distribution approaches the true posterior.
>
> > Could you explain the second equality in the derivation leading to eq. 26? For example, by a step-by-step derivation.
>
> We utilize equality between the variational and negative distribution to simplify the denominator. We have included the following text and additional line to make the derivation clearer
>
> > using $p^-(\\boldsymbol{\\theta})=q\_{\\boldsymbol{\\phi}}(\\boldsymbol{\\theta})$, we have, $\\sum\_{k=1}^K  q\_{\\boldsymbol{\\phi}}(\\boldsymbol{\\theta}\_{k})/p^-(\\boldsymbol{\\theta\_{k}})=K$, giving
> $$
> = \\frac{\\nabla\_{\\boldsymbol{\\phi}} \\sum\_{k=1}^K q\_{\\boldsymbol{\\phi}}(\\boldsymbol{\\theta}\_{k})/p^-(\\boldsymbol{\\theta}\_{k})}{K},
> $$
>
> > Is there a condition that the posterior is contained within the parametric variational family missing?
>
> Yes, we have now added the following to section 2.2.
>
> > Assume that there exists a $\\boldsymbol{\\phi}$ such that $p(\\boldsymbol{\\theta}|\\mathbf{x}\_\\text{obs})=q\_{\\boldsymbol{\\phi}}(\\boldsymbol{\\theta})$.
>
> Thank you again for your review and eye for detail that has helped us improve the paper.
>
> [1] Yao, Yuling, et al. "Yes, but did it work?: Evaluating variational inference." International Conference on Machine Learning. PMLR, 2018.

---

> > ### Comment · Reviewer_36B5 · 2024-11-25
> > **Responds**
> >
> > The rebuttal effectively addresses my concerns regarding the paper. I have reviewed the updated version and confirmed that it incorporates the changes as stated. Consequently, I will raise the Presentation score to 4. I have no further questions at this time.

---

### Official Review · Reviewer_SFto · 2024-11-01

**Soundness:** 4
**Presentation:** 2
**Contribution:** 3
**Rating:** 10
**Confidence:** 4

**Summary:**

The paper proposes a new way to approximate the posterior distribution over parameters for a parametric model $p(x;\theta)$.

The approach:
* defines 2 multinomial distributions:
   *  $y(\\{\theta_k\\})$ over the index $k$ of $K$ parameter samples $\theta_k$ as a function of the model $p(x,\theta)$ and noise distribution;
   * $\hat{y}(\\{\theta_k\\})$ as a function of the parameterised approximate posterior $q_\phi(\theta)$ and the noise distribution;
* $\hat{y}$ is trained to match $y$ under cross entropy loss;
* at optimality $q_\phi(\theta)$ matches the true posterior $p(\theta|x)$ (subject to the choice of the parametric family $q_\phi$).

Results compare the proposed method to typical amortised variational inference (maximising the ELBO) and a more similar method, showing improved outcomes on those datasets.

**Strengths:**

* The proposed method makes sense and appears straight-forward to implement.
* The method appears general to problems with a parametric model where $p(x,\theta)$ can be computed.
* Comparison to SNIS is revealing and clear.

**Weaknesses:**

Overall I like the paper, find it to be of high quality and soundness and the proposed method is a clever trick that converts a difficult problem into a simpler classification-type task. Areas in which the paper could improve (from good to excellent) are:

**Clarity**: as someone familiar with this field but not immersed in it, the paper could be clearer to read. As specific examples:
* [033, 078] From the outset, the paper refers to *latent variables*, which often refer to variables that are one-to-one with data samples, whereas the term *parameters* typically relates to global variables of the model, i.e. common across all samples. (Of course parameters are often treated as unobserved, hence latent, variables). I appreciate that this may be subjective, but I find it to be fairly universal and a useful distinction, so this phrase put the wrong starting point in my mind. I was wondering throughout when "parameters of the model" get updated and only when I reached the experiments did it become clear that it is about parameters and it made more sense. I would suggest referring to "parameters" (treated as latent variables).
* The preamble to Eq 2 is not directly relevant and could be more clear: labels y are constructed analytically and the approximate posterior is trained by learning to predict them. The backstory of predicting k* doesn't seem so relevant (as it is in NCE, say) so this bit threw me off until I realised that.
* [202-213] This part stands out as less precise relative to preceding arguments. The method trains one ratio of ratios to match another ratio of ratios using cross entropy loss. Reference to "classification being too easy" and "punishing" seems out of place and unclear. The issue seems to be the relative range of values in the softmax function (+/- numerical instability?), which is mentioned but blended with loose wording. This part could be improved, particularly as computing these ratios/choosing the negative sampling distribution is the crux and potential pitfall of ratio estimation.
* [161] Ensuring $\pi$ covers the true posterior - I can't see how this is "guaranteed", unless pi is diffuse, which sounds problematic for ratios.
* Experiments: to compute a posterior, you need a prior, but the priors don't seem clearly specified.
* Fig 1: how are some log probability values positive?
* GARCH experiment: The descriptions of $\alpha_1$ and $\beta_1$ are rather vague.

**Analysis/Experiments**: For such a widely applicable method, the experiments to showcase its efficacy/analyse performance etc seem thin.
* Understanding how well it works in higher dimensions is of interest, where potentially finding a negative distribution close to the true posterior could be less straightforward.
* Understanding how the methods compare as number of data points increases is of interest to know when the method is most applicable.
* Understanding how the methods compare when the posterior is mis-specified is of interest to understand robustness (mode seeking may be preferable to distribution covering?).

Minor
* [134] it would aid clarity to explicitly say optimisation is "with respect to $\phi$"
* [134] it would be much clearer to state labels y "(from Eq 2)".
* [370] "a different observation" of what?
* [421-424] the paper does well to be relatively self-contained, this part could be made more clear to the less familiar reader.

**Questions:**

* 50k "optimisation steps": this sounds a lot, any explanation for this? What is the definition of step?
* K=8: Any ablation of the effect of K on accuracy vs wall-clock time? i.e. trade off in slower steps vs number of steps
     * Fig 6 (K=2) is hard to compare to Fig 1 (K=8) due to their separation and the SNIS result dominates the scale, squashing up the rest.

---

> ### Author Response · Authors · 2024-11-25
>
> We are grateful for your positive remarks about the paper, and we will incorporate your useful suggestions as best we can. In the updated manuscript, we have avoided using the terminology "latent" in the updated paper due to the potential confusion you mentioned. We agree that there was some loose wording in place and have tightened this up.
>
> > [202-213] This part stands out as less precise relative to preceding arguments..."
>
> We have removed the informal references to "classification problem becoming too easy", and avoided using the term "punish". Whilst considering the relative range of the softmax inputs is a factor as you suggest, we found partitioning the loss and considering the ratio sum to be beneficial to build intuition, particularly for why a lower value of $\alpha$ leads to more mass-covering behaviour. We do note that as mentioned in the conclusion, further work, both empirically, and theoretically would be beneficial for guiding the choice of negative distribution.
>
> > Ensuring covers the true posterior - I can't see how this is "guaranteed", unless pi is diffuse, which sounds problematic for ratios.
>
> In this section, we are referring to ensuring the supports of the distributions are compatible, which is required to recover the true posterior at optimality of the classifier (under the conditions listed in section 2.2). This is not a claim about whether optimization will lead to the true posterior in practice. For example, consider a bimodal posterior which we assume $q$ has the sufficient flexibility to model. If at initialization negligible mass is placed on one of these modes, we are likely to miss that mode, and optimize to a local minimum (like any other variational inference method). We note that the language we used - "In practice" - was potentially misleading in this context, so we have removed this in the updated manuscript.
>
> > Fig 1: how are some log probability values positive?
>
> These are continuous distributions (probability densities). If your concern is that we do not specify log densities, we will note our use aligns with other work [1].
>
> > Understanding how well it works in higher dimensions is of interest, where potentially finding a negative distribution close to the true posterior could be less straightforward.
>
> In the updated paper, we include experimental results with Bayesian neural networks with dimensionality up to $\boldsymbol{\theta} \in \mathbb{R}^{3000}$, and show very similar, but marginally worse performance than using the ELBO (see appendix A5).
>
> > Understanding how the methods compare when the posterior is misspecified is of interest to understand robustness (mode seeking may be preferable to distribution covering?).
>
> We note that although we do not consider purposely misspecified posteriors, all the tasks considered will be misspecified to some degree. This will be particularly the case for the Bayesian neural network example. Note, that despite being a more mass covering objective, SoftCVI still leads to similar predictive performance.
>
> > 50k "optimisation steps": this sounds a lot, any explanation for this? What is the definition of step?
>
> A step refers to $N$ in algorithm 1. 50,000 is likely overkill in general, but we found it to work reasonably across all the methods, and we would prefer the results to reflect the results of the approaches after they have approximately converged. If you tuned the learning rate for each objective and task, we would expect you could get similar results with significantly less steps.
>
> > K=8: Any ablation of the effect of K on accuracy vs wall-clock time? i.e. trade off in slower steps vs number of steps.
>
> We have included an additional figure Fig. 7, plotting the accuracy as a function of $K$ for each method, in addition to the associated run times. In general, we found $K$ to increase performance, but to increase runtime approximately linearly on CPU.
>
> [1] Lueckmann, Jan-Matthis, et al. "Benchmarking simulation-based inference." International conference on artificial intelligence and statistics. PMLR, 2021.

---

> ### Comment · Reviewer_SFto · 2024-12-02
>
> The authors addressed well the points I raised and I think the paper has improved to "strong accept, should be highlighted at the conference".  Based on that description, I have raised my score: 8 $\to$ 10.
>
> To expand slightly, I don't particularly like the ICLR scoring system as 10/10 sounds a bit too "perfect". There is no doubt room for improvement with the proposed approach and the larger experiments added in the rebuttal show the ELBO "fighting back" (e.g. the MNIST images look more diverse and results in Table 1). But the proposed method is theoretically grounded, may be useful "as is" for smaller scale problems and may spur further advances for larger scale problems.
>
> (Note to authors for future: it would make life easier for reviewers if main manuscript changes were highlighted, even though not asked for)

---

> > ### Author Response · Authors · 2024-12-04
> >
> > We’re glad the revisions addressed your concerns and appreciate you updating your score. We apologize for not highlighting the changes and will ensure to do so in the future.

---

### Official Review · Reviewer_xUmn · 2024-11-02

**Soundness:** 2
**Presentation:** 3
**Contribution:** 2
**Rating:** 5
**Confidence:** 4

**Summary:**

The paper considers a variational inference approach that builds upon classifying ‘noisy’ samples from a learned proposal distribution as negative samples, similar to a noise contrastive estimation approach, and where ‘ground truth soft labels’ depend on a self-normalised importance sampling estimate for the log ratio of the target density relative to the proposal density. In contrast to a classical ELBO minimizing the reverse KL to the target, the suggested approach appears to better approximate the forward KL to the target. Numerical experiments on benchmark Bayesian inference problems illustrate that suggested approach yields te better coverage probabilities and yields to approximations with less bias for the posterior mean estimate.

**Strengths:**

The paper addresses the classic and important question of Bayesian inference, particularly for the challenging setting where the approximate posterior should be close to the target with respect to the forward KL to yield well calibrated inferences. The approach avoids previous objectives that yield mass-covering behaviour, but are more challenging to optimise, particular in high dimensions. The suggested objective is now as far as I am aware. It is also very interesting that the suggested approach can be used to construct a previously constructed objective (SNIS-fKL), but with a gradient estimator that has zero variance in the idealised setting when the approximate posterior coincides with the true posterior. Experiments do indeed show that the suggested method improves the coverage and leads to approximate posterior that are closer to the target in terms of the forward KL.

**Weaknesses:**

The empirical verification seems a bit limited. In particular, the only baseline (beyond a standard ELBO) is the SNIS-fKL objective. However, as the paper shows, they have the same expected gradients. It is thus not clear how well the suggested approach compares to other (approximate) Bayesian methods that have different averaged gradients, e.g. [1-3].

Also as the suggested approach has a higher computation complexity requiring sampling and density-evaluations of K particles, evaluations with a simple IWAE approach instead of the ELBO baseline appears appropriate.

The authors consider (in Section 2.1) that they have access to samples from the true posterior. However, is having access to samples from the true posterior not the aim of the method? In (4), is $y$ equal to $\hat{y}$ as in Algorithm 1 when one does not have true posterior samples?

It is not clear why eq. (4) is the right objective to optimise when one does not have access to the true labels $y$. Does optimizing it correspond to some proper scoring rule or divergence [4] that are minimized for this objective?

Can you clarify why the optimal classifier recovers the true posterior and that the true posterior is equal to the approximate variational distribution as per eq. (6)? The ‘optimal classifier parameterized as in eq. (3)’ depends (apparently?) on the number of samples $K$ from the proposal and on the proposal distribution. Do all the arguments on page 3 from line 141 not depend on these choices? I understand that the classifier is optimal with respect to a cross-entropy loss if it recovers the Bayesian posterior [4]. But is this optimality even realisable in the suggested form (3)?

The experiments target posteriors that are not that high-dimensional (<=200). Does the method scale to high-dimensional problems? Does it overcome scalability issues commonly faced when optimizing the forward KL or for importance sampling?


[1] Dieng, Adji Bousso, et al. "Variational Inference via $\chi $ Upper Bound Minimization." Advances in Neural Information Processing Systems 30 (2017).\
[2] Li, Chengrui, et al. "Forward $\chi^ 2$ Divergence Based Variational Importance Sampling." arXiv preprint arXiv:2311.02516 (2023).\
[3] Naesseth, Christian, Fredrik Lindsten, and David Blei. "Markovian score climbing: Variational inference with KL (p|| q)." Advances in Neural Information Processing Systems 33 (2020): 15499-15510.\
[4] Mohamed, Shakir, and Balaji Lakshminarayanan. "Learning in implicit generative models." arXiv preprint arXiv:1610.03483(2016).

**Questions:**

It would be helpful if the paper is a bit clearer what kind of reference posteriors are used. Are they from MCMC algorithms that can be assumed to closely approximate the true posterior? What about the SBI posteriors?

Re line 127, can you elaborate what is the exact parameterisation of the classifier in terms of the ratio of two distributions. Are the parameters (infinite-dimensional) probability densities?

Can you clarify why you compare against simulation-based reference models (SBI)? How is the proposed approach applicable without having access to the likelihood function?

---

> ### Author Response · Authors · 2024-11-25
> **Part 1 (split due to character limit)**
>
> Thank you for your thoughtful review and suggestions. We chose to compare to the ELBO as it is the most widely used and successful variational objective, and SNIS-fKL as 1) it is suggested as an approach that encourages mass-covering behaviour, 2) it shows competitive performance with other mass-covering objectives [e.g. 1, 2], and 3) it is closely related to a special case of SoftCVI. Note, we did have initial experiments with the IWAE objective as implemented in the original paper [3], which performed poorly, but we felt it was not a fair comparison to include without further investigation of alternative gradient estimators [4].
>
> > Also as the suggested approach has a higher computation complexity
>
> In the updated version we report more comprehensive runtimes as a function of $K$ (Fig. 7). In general, the run times are comparable or faster for SoftCVI or SNIS-fKL compared to the ELBO when using the same number of particles (on the tasks we consider). Note that it is important to consider the speed of gradient computation. The ELBO differentiates through the sampling process using reparametrization gradients, whereas SoftCVI does not. This can lead to differences in run times, particularly when there is asymmetry between the sampling speed and density evaluation speed (e.g. SLCP example, which uses a masked autoregressive flow). Overall, the results are architecture specific - the ELBO would have likely had lower run times on the SLCP task had we chosen to use an inverse autoregressive flow [5]. We have mentioned this in the figure caption.
>
> There were a couple of questions suggesting that we require access to true posterior samples, which we will address together
>
> > The authors consider (in Section 2.1) that they have access to samples from the true posterior. However, is having access to samples from the true posterior not the aim of the method?
>
> > In (4), is $y$ equal to $\hat{y}$ as in Algorithm 1 when one does not have true posterior samples?
>
> SoftCVI never uses or relies on having access to samples from the true posterior. We understand the framing could have been clearer in the original manuscript and have updated section 2.1 in the new version to address this. Specifically, in the updated section 2.2, we avoid considering having access to the true posterior samples, which seemed to be a source of confusion for reviewers. To provide some clarity, the ground truth soft labels $\mathbf{y}$ are available to compute for any set of samples $\\{\boldsymbol{\theta}_k\\}\_{k=1}^K$, using equation 2. To further make this clearer, we have also updated the following paragraph in section 2.1 to explicitly reference equation 2:
>
> > Nonetheless, by instead generating samples from a proposal distribution,  $\\{\boldsymbol{\theta}_k\\}\_{k=1}^K \sim \pi(\boldsymbol{\theta})$, and by
> assigning soft labels to these samples using eq. (2),  it is still possible to train a classifier which at optimality recovers the true posterior, as we will discuss in the subsequent section.
>
> There were a couple of questions regarding the correctness of SoftCVI, which again, we will address together
>
> > It is not clear why eq. (4) is the right objective to optimise when one does not have access to the true labels. Does optimizing it correspond to some proper scoring rule or divergence [4] that are minimized for this objective?
>
> > Can you clarify why the optimal classifier recovers the true posterior and that the true posterior is equal to the approximate variational distribution as per eq. (6)?
>
> As mentioned in response to the previous comments, we can compute the true labels using eq. (2). We are minimizing the cross entropy between the true and predicted labels, and we have shown in section 2.2 that the optimal classifier, minimizing the cross entropy for all $\\{\boldsymbol{\theta}_k\\}\_{k=1}^K \in \Theta^K$, recovers the true posterior (under the conditions listed). Evidently, we also discuss the connections to the SNIS-fKL divergence, which provides further merit to the correctness of our method.
>
> To provide some intuition, the optimal classifier must learn the density ratio between the positive and negative distributions (up to a constant). If we choose a classification problem such that the positive distribution is the true posterior (up to a constant), and train a classifier parametrized as $q_{\boldsymbol{\phi}}(\boldsymbol{\theta})/p^-(\boldsymbol{\theta})$, this leads to the optimal classifier being the posterior (under certain conditions). This result does not rely on any specific value of $K$; we can also see this in appendix A1, regardless of the choice of $K$, the gradient will be zero when the true posterior is recovered for any set $\\{\boldsymbol{\theta}_k\\}\_{k=1}^K \in \Theta^K$. Note that in practice, the choice of $K$ can impact performance, due to the various limitations of variational inference, such as misspecified variational distributions, and optimization only finding a local minimum.

---

> > ### Author Response · Authors · 2024-11-25
> > **Part 2**
> >
> > > The ‘optimal classifier parameterized as in eq. (3)’ depends (apparently?) on the number of samples from the proposal and on the proposal distribution.
> >
> > The optimal classifier does not depend on the proposal distribution (see eq 2). Regardless of $K$ or the proposal distribution (assuming the necessary conditions from section 2.2 are met), the optimal classifier learns the same ratio between the positive and negative distributions. $K$ evidently needs to be considered when generating labels, due to the fact we need to normalize over the set of $K$ samples under consideration (see also our response to reviewer AxW5).
> >
> > > It would be helpful if the paper is a bit clearer what kind of reference posteriors are used. Are they from MCMC algorithms that can be assumed to closely approximate the true posterior? What about the SBI posteriors?
> >
> > We have added more detailed task descriptions and the source of the reference samples in appendix A7. The SLCP posterior was obtained using  sampling/importance resampling with the analytical likelihood function (not SBI), the linear regression task analytically, and the others tasks using MCMC. These posteriors are from packages designed for reliable benchmarking purposes for comparison of Bayesian inference methods.
> >
> > > Re line 127, can you elaborate what is the exact parameterization of the classifier in terms of the ratio of two distributions. Are the parameters (infinite-dimensional) probability densities?
> >
> > The classifier is parameterized as described in eq. 3 (not infinite-dimensional probability densities). If you are referring to the exact parameterization of the variational distribution $q_{\phi}$, we have given a description with each task, and have provided code which will invariably be the most precise way to share the parameterizations of the variational distributions.
> >
> > > Can you clarify why you compare against simulation-based reference models (SBI)? How is the proposed approach applicable without having access to the likelihood function?
> >
> > The SLCP task has a tractable (Gaussian) likelihood function, despite its common use in the SBI literature. SoftCVI requires access to the likelihood function.
> >
> > > The experiments target posteriors that are not that high-dimensional (<=200). Does the method scale to high-dimensional problems?
> >
> > We agree that investigation of higher dimensional problems is important. In the updated manuscript, we consider applications to VAEs (appendix A4) and Bayesian neural networks (appendix A5). In both cases, the results suggest the method is competitive with the ELBO.
> >
> > In light of these clarifications to the problem setup, in addition to the additional experimental results, we kindly ask you to reconsider your rating.
> >
> > [1] Glöckler, Manuel, Michael Deistler, and Jakob H. Macke. "Variational methods for simulation-based inference." arXiv preprint arXiv:2203.04176 (2022).
> >
> > [2] Jerfel, Ghassen, et al. "Variational refinement for importance sampling using the forward kullback-leibler divergence." Uncertainty in Artificial Intelligence. PMLR, 2021.
> >
> > [3] Burda, Yuri, Roger Grosse, and Ruslan Salakhutdinov. "Importance weighted autoencoders." arXiv preprint arXiv:1509.00519 (2015).
> >
> > [4] Tucker, George, et al. "Doubly reparameterized gradient estimators for monte carlo objectives." arXiv preprint arXiv:1810.04152 (2018).
> >
> > [5] Kingma, Durk P., et al. "Improved variational inference with inverse autoregressive flow." Advances in neural information processing systems 29 (2016).

---

> > > ### Comment · Reviewer_xUmn · 2024-11-26
> > > **Response to rebuttal 2**
> > >
> > > I thank the authors for providing some initial results on the application of their method to a VAE model for MNIST that illustrate that their approach is competitive with the ELBO. Can you clarify how to sample $\theta_k$ in eq. (27) for the newly introduced objective that involves the generative model part, and how to compute the gradients (as mentioned without relying on reparameterized gradients).

---

> > > > ### Author Response · Authors · 2024-11-26
> > > >
> > > > The SoftCVI objective does not utilize repararameterization gradients. This is likely most clear from the framing in algorithm 1, noticing that the sampling is performed separately to the loss gradient computation. The addition of the model component in eq. (27) does not change this. We have added the following sentence after eq. (27) to make the source of the samples clearer:
> > > >
> > > > > where in the above formulation we reuse the proposal distribution samples $\\{\\boldsymbol{\\theta}\_k\\}\_{k=1}^K$ already sampled for the SoftCVI objective, and we assume the proposal distribution is equal to the variational distribution.
> > > >
> > > > In terms of implementation, we can either carry out the sampling upfront prior to gradient computation, or we can carry out the sampling within the objective, but utilize a stop gradient operator for the sampling operation to ensure correct gradients.

---

> > ### Comment · Reviewer_xUmn · 2024-11-26
> > **Response to rebuttal**
> >
> > I thank the authors for their response that has clarified some points. However, it has still not become clear to me how all the arguments leading to eq. (6) provide some justification of the method. The authors assume in line 141 that eq. (6) holds for some $\phi$, i.e. there is some unknown parameter of the variational/proposal density that recovers the true posterior. But even if this holds, how does optimising the suggested objective then imply that I can learn this $\phi$. Does this even hold for $K=1$ (as the presented arguments seem to not use $K$)? As far as I understand, the stationarity of the loss function for any $\theta$ presented later in appendix A1 only holds when the optimal classifier is obtained, i.e. when the (learned) variational distribution equals the posterior.

---

> > > ### Author Response · Authors · 2024-11-26
> > >
> > > Thank you for your response.
> > >
> > > > The authors assume in line 141 that eq. (6) holds for some $\phi$, i.e. there is some unknown parameter of the variational/proposal density that recovers the true posterior.
> > >
> > > Note that the condition that $p(\\boldsymbol{\\theta}|\\mathbf{x}\_{\\text{obs}})=q\_{\\boldsymbol{\\phi}}(\\boldsymbol{\\theta})$ for some $\\boldsymbol{\\phi}$ is necessary for all variational inference methods when considering the conditions required for the true posterior to be recovered. Just to be clear, this condition is unlikely to hold in practice, but in section 2.2 we are considering idealised conditions.
> > >
> > > >  But even if this holds, how does optimising the suggested objective then imply that I can learn this $\phi$.
> > >
> > > The principle that a classifier minimizing the softmax (or binary) cross-entropy objective recovers the density ratio between positive and negative distributions up to a constant, has been well established in the literature (given assumptions such as sufficient data and a sufficiently flexible classifer). For instance, in the case where $K=2$, this is discussed in section 2.1 of [1], while for $K\geq2$, a similar result is presented and relied on in section 3 of [2].  If you have concerns about this point, we would be happy to clarify further. The equations leading up to equation 6 then follow directly from this. Given this, section 2.2 demonstrates that under the conditions listed, minimizing the objective for all $\\{ \\boldsymbol{\\theta}\_k \\}\_{k=1}^K \\in \\Theta^K$, recovers the $\\boldsymbol{\\phi}$ value corresponding to the true posterior.
> > >
> > > > Does this even hold for $K=1$ (as the presented arguments seem to not use $K$)?
> > >
> > > No, we need $K\geq2$. Setting $K=1$ does not lead to a valid classification problem for optimisation.  The method relies on contrasting samples in order to learn the density ratio, which requires at least two samples. We can see this in equations 2 and 3, setting $K=1$ leads to a scalar input into softmax, which will return one regardless of the value of the log ratios, and as such learning is not possible.
> > >
> > > > As far as I understand, the stationarity of the loss function for any $\theta$ presented later in appendix A1 only holds when the optimal classifier is obtained, i.e. when the (learned) variational distribution equals the posterior.
> > >
> > > This is correct and desirable, i.e. there is a stationary point in the loss function due to the loss being minimized when the variational distribution equals the true posterior.
> > >
> > > We hope that this has clarified some points, and if you feel we have adequately addressed your concerns, we would appreciate if you could reconsider your rating. Let us know if you have any further questions.
> > >
> > > [1] Gutmann, Michael U., and Aapo Hyvärinen. "Noise-contrastive estimation of unnormalized statistical models, with applications to natural image statistics." Journal of machine learning research 13.2 (2012).
> > >
> > > [2] Durkan, Conor, Iain Murray, and George Papamakarios. "On contrastive learning for likelihood-free inference." International conference on machine learning. PMLR, 2020.

---

> ### Comment · Reviewer_xUmn · 2024-12-03
> **Response**
>
> I thank the authors for providing some context to the references [1-2]. If the correctness of their approach follows from well-established previous woks, I would encourage the authors to make theses connections more precise (e.g. with respect to K, and how K affects any bias).
>
> Regarding the implementation for the VAE model, I encourage to authors to provide more details how the proposal distribution is optimised (e.g. how this works without reparameterization, via stop gradients, does this give unbiased gradients?) to ensure that the proposal distribution becomes close to the evolving posterior during training (otherwise, I don't see how L_model optimizes well the marginal log-likelihood).

---

> > ### Author Response · Authors · 2024-12-04
> >
> > Thank you for your response. Although the revision period has passed, we note that we discuss the relationship with other work in Section 3.3, where we introduce InfoNCE and simulation-based methods, which similarly use an approximate posterior as a component of a classifier. Regarding bias, I presume you are referring to bias in the gradients if we treat the gradient estimator as an estimator of the forward KL divergence gradient? In the case of $\alpha=1$, we have from the equivalence to SNIS-fKL that the bias decreases with $\mathcal{O}(1/K)$. More generally, the gradient bias converges to zero as the variational distribution approaches the true posterior, although we note this property holds for any objective where the expected gradient becomes zero when the posterior is perfectly recovered. While there is undoubtedly room for further theoretical work, we hope our primary contributions are clear: the novel framing of variational inference as a classification problem, a theoretical comparison to SNIS-fKL, and demonstrating improved empirical results.
> >
> > For the VAE model and all other experiments in the paper, the proposal distribution is chosen to equal the variational distribution, and the updates are performed identically across the experiments. The approach used is in algorithm 1, notice that there is no need for reparameterized gradients  - we compute the gradient of a function which does not even include a sampling operation. We can utilize a stop gradient operator and include sampling within the objective, but this is an implementation detail that results in gradients equivalent to those obtained using Algorithm 1.

---

### Official Review · Reviewer_AxW5 · 2024-11-02

**Soundness:** 3
**Presentation:** 3
**Contribution:** 3
**Rating:** 6
**Confidence:** 4

**Summary:**

In this paper, the authors propose an approach that reformulates the inference problem as a classification task using contrastive learning and the generation of soft labels.

These soft labels, derived from an unnormalized target distribution, are employed in a cross-entropy loss to fit the variational distribution. The authors demonstrate the efficacy of their methodology through a series of experiments on Bayesian inference.

**Strengths:**

The paper presents an elegant method for fitting the variational distribution by training the model to solve a classification problem.

I identify two key benefits of the proposed approach:

1. There is no need for samples from either the negative or the target distribution.
2. The densities of the target and negative distributions need only be known up to a normalizing constant.

Overall, the paper is well-written and easy to follow.

**Weaknesses:**

I note the following weaknesses in the paper:

### Major:

1. Deep Learning examples:

The experimental evaluation feels modest and misses an opportunity to show the benefits of the proposed methodology in deep learning scenarios. The authors focus solely on "classical" examples of Bayesian inference. I think it would be very illustrative to consider deep learning applications.

For example, in the related work section, the authors mention papers on Variational Auto-Encoders (VAE). It would be interesting to see how the proposed methodology performs in this context.

I see several ways your approach could be applied:

- End-to-End VAE training: Instead of optimizing the standard ELBO (or alternatives like IWAE [1]), incorporate the SoftCVI training procedure and evaluate the negative log-likelihood.
- Fitting variational distribution for pretrained VAE: For an already pretrained VAE (decoder) with a complex, multimodal posterior, apply your training procedure to fit a variational distribution to this posterior. Then demonstrate that the variational distribution fitted with SoftCVI covers more modes and provides more diverse samples (see e.g. [2]).

I appreciate the example with the SLCP task. Similarly, an example with a "circular" posterior in [3] could be considered for demonstration.

2. Training curves with different $K$:

Since $K$ is a hyperparameter we can choose as we wish, it's important to understand how it influences the results. In Figures 1 and 6, two different choices of $K$ are considered, but there is no discussion on which is better. It would be beneficial to see not only the final evaluations but also the training curves for different $K$ values. Given that $K$ can range widely (from 2 to infinity), practical guidance on selecting $K$ would be valuable.

Additionally, training curves are interesting in the context of comparison with SNIS-fKL, as the variance of estimation will be greater there, and it is interesting to see it.

### Minor:

1. Motivation in Section 2.1:

Initially, I found the beginning of Section 2.1 a bit confusing. It mentions a scenario with $K−1$ negative samples and only 1 positive, which doesn't seem to match the scenario in the paper. As I understand it, we have samples only from the proposal distribution and assign probabilities to them as if they come from the true distribution to generate soft labels. We don't assume that only one sample comes from the true distribution. If my understanding is incorrect, please clarify. Otherwise, consider rephrasing this section for clarity.


2. Typo:

Line 329: "in contrast **to to**".

----

I am positive about the methodological contributions. With the incorporation of extensions to the experimental section, I am willing to significantly increase my evaluation score.


----

References:

[1] Burda, Y., Grosse, R., & Salakhutdinov, R. (2015). Importance Weighted Autoencoders. arXiv preprint arXiv:1509.00519.

[2] Thin, A., Kotelevskii, N., Denain, J. S., Grinsztajn, L., Durmus, A., Panov, M., & Moulines, E. (2020). MetFlow: A New Efficient Method for Bridging the Gap Between Markov Chain Monte Carlo and Variational Inference. arXiv preprint arXiv:2002.12253.

[3] Thin, A., Kotelevskii, N., Doucet, A., Durmus, A., Moulines, E., & Panov, M. (2021). Monte Carlo Variational Auto-Encoders. In International Conference on Machine Learning (pp. 10247-10257). PMLR.

**Questions:**

1. In line 57, the authors mention proposing **a family of variational objectives**. Could you clarify this point? It seems that there is only one objective—the cross-entropy between true soft labels and the predicted categorical distribution. Is there indeed a family of objectives, or is it a single objective?

2. [questions from Weaknesses] How does the choice of $K$ influence the training curves and final results? Is there an optimal value for $K$? What criteria should be used to select $K$?

---

> ### Author Response · Authors · 2024-11-25
>
> We appreciate your positive comments on the benefits our our approach and your helpful suggestions for additional experiments. Based on your comments, we have added two deep learning examples, training VAEs, and fitting of Bayesian neural networks, in appendices A.4 and A.5, respectively.
>
> >  How does the choice of $K$ influence the training curves and final results? Is there an optimal value for $K$ What criteria should be used to select $K$.
>
> We have included a figure in the appendix, showing the log prob. $\boldsymbol{\theta}^*$ metric for different values of $K$ for all tasks, to provide better guidance for the choice of $K$. In general, we found higher values of $K$ to perform better, and as such we have highlighted this in the results by adding the following sentence
>
> >  Both SoftCVI and SNIS-fKL tended to place more mass on the reference samples when trained with a larger $K$, but this comes with an increase in computational cost (fig. 7).
>
> We found that plotting training curves based on the loss values does not provide a useful comparison, due to the different scales for the different objectives. Particularly for SoftCVI, the classification problem generally becomes more challenging as training progresses, which can lead to non-decreasing losses despite the variational distribution improving. It is likely possible to compute and track other metrics throughout training (e.g. log prob. $\boldsymbol{\theta}^*$). We will aim to include this if we have the time.
>
> > the beginning of Section 2.1 a bit confusing
>
> We have revised section 2.1 to make it clearer. Specifically, we have avoided considering the case where we have access to true posterior and negative distribution samples, which seemed to be a source of confusion. Additionally, we have avoided the notation of $k^*$ for the true sample, which is also less important compared to e.g. NCE. Note however, that the probabilities, $\mathbf{y}$, are assigned under an assumed classification problem in which there exists a (single) true sample in the set of $K$ samples. It is easy to see why this distinction is needed by a counter example; if we instead assumed e.g. all $K$ samples were true samples, the optimal classifier would be $y_k=1,\ k=1,...,K$ (and learning would not be possible). Throughout the paper we make the convenient choice of assigning probabilities under the assumed classification problem with a single positive example, which leads to a simple cross-entropy objective.
>
> > In line 57, the authors mention proposing a family of variational objectives. Could you clarify this point?
>
> We are referring to different choices of $\alpha$, or negative distributions more generally, leading to different objective functions with different properties.
>
> Thank you again for your review and suggestions. We hope that with the additional experimental results we have addressed your primary concerns and that you consider adjusting your rating as you deem appropriate.

---

> ### Comment · Reviewer_AxW5 · 2024-11-26
>
> Dear Authors,
>
> Thank you for your thorough responses and the changes you've incorporated into the manuscript. I appreciate your efforts to address my concerns.
>
> For future reference, I kindly request that you specifically highlight the parts you've changed, perhaps using a different color or highlighting. For example, in Section 2.1, since everything is still written in black, it's difficult to see exactly what has been modified.
>
> ----
>
> I am willing to increase my evaluation score to reflect my appreciation of your efforts. However, I still have a concern (or perhaps I'm misunderstanding) about the interpretation of having $K - 1$ samples and one single sample from the true posterior.
>
> Perhaps we are discussing the same concept from different perspectives. Allow me to clarify my viewpoint:
>
> I consider your objective as follows: You have two distributions. The first is the categorical "ground truth," parameterized via the joint distribution (Equation 2). The second is the categorical "predicted" distribution, parameterized via the variational distribution (Equation 3). You use Cross-Entropy (CE), which is a proper scoring rule, to fit the second distribution to match the first. Because CE is proper, minimizing this objective ensures that the variational distribution matches the joint distribution.
>
> Therefore, I don't see why it's important to emphasize that $K−1$ samples are "artificial" and only one comes from the ground truth. From my perspective, your objective in Equation 7 can be viewed as a variational inference (VI) objective, specifically minimizing the Kullback-Leibler (KL) divergence between the "ground truth" and "predicted" distributions. The CE between the "ground truth" and "predicted" distributions can be decomposed into the entropy of the "ground truth" distribution plus the KL divergence between the "ground truth" and "predicted" distributions. Since there is no gradient through the entropy term, your objective is equivalent to minimizing the KL divergence, which might be easier to understand in the context of VI.
>
> What concerns me about interpreting only one sample as coming from the ground truth is that the ideal solution would provide you with $K$ samples from the "ground truth". Therefore, it seems confusing to suggest that only one sample comes from the ground truth.
>
> Please let me know if this makes sense.

---

> > ### Author Response · Authors · 2024-11-26
> >
> > Thank you for the reconsideration of your score and the interesting discussion. From your description, it seems that you have the correct interpretation of the method, but are viewing it slightly differently.
> >
> > The key issue here is understanding what exactly the ground truth labels (or the categorical distribution event probabilities, if you prefer), represent. Intuitively, computing the labels involves computing the ratio $p(\\boldsymbol{\\theta}, \\mathbf{x}\_{\text{obs}}) / p^-(\\boldsymbol{\\theta})$ for each of the $K$ samples. The ratio quantifies the relative likelihood of each sample being drawn from the true posterior, compared to the negative distribution. Using equation 2, we can convert these into a set of categorical event probabilities, where each individual probability represents the probability of the corresponding sample being the single true sample in the assumed classification problem. Without this classification framing, it seems less clear what the ground truth labels represent.
> >
> > Your subsequent observation on the objective, regarding minimizing the forward Kullback-Leibler (KL) divergence between the ground truth and predicted categorical distributions, we agree with, and this is consistent with our framing.
> >
> > > What concerns me about interpreting only one sample as coming from the ground truth is that the ideal solution would provide you with $K$ samples from the "ground truth".
> >
> > Maybe it will help to consider an example optimal scenario. For example, if we select $p^-(\\boldsymbol{\\theta})=\pi(\\boldsymbol{\\theta})=q_{\\boldsymbol{\\phi}}(\\boldsymbol{\\theta})$, then as the variational distribution approaches the posterior, the positive and negative distribution will approach equality. This will result in the ground truth labels being $1/K$, i.e. all the samples are deemed equally likely to be the single true posterior sample under the assumed classification problem. This is logically consistent - if the negative and positive distributions are considered equal, then there will be no possibility of meaningful distinction between the samples. As such the optimal classifier should also assign equal probabilities to all the samples, which is achieved by $q\_{\\boldsymbol{\\phi}}(\\boldsymbol{\\theta})$ being equal to the true posterior.

---

### Comment · Area_Chair_1HS1 · 2024-12-02
**Discussions between reviewers and authors**

Time for discussions as author feedback is in. I encourage all the reviewers to reply. You should treat the paper that you're reviewing in the same way as you'd like your submission to be treated :)

---

### Meta-Review · Area_Chair_1HS1 · 2024-12-17

**Metareview:**

This paper proposes a noise contrastive estimation (NCE) type of objective for variational inference. The idea is to use the unnormalised posterior and a noise distribution to assign a "soft label" (i.e., the classification probability vector) on samples from a proposal distribution, and then fit another classifier to this "dataset" where the classifier itself is defined using density ratios between an approximate posterior and the same noise distribution used in the "soft label" generation process.
The idea is clear and the presentation is sound for people who knows variational inference and NCE. Initially reviewers had questions regarding some details, and they also requested further experiments. These concerns were mostly addressed during author rebuttal.

After a brief read, I was a bit surprised that, although the idea looks very obvious in hindsight, to the best of my knowledge no previous work has formalised this idea and made it work. Multi-class NCE has been around for sometime regarding mutual info estimation but it doesn't work very well in high dimensions, so I would be curious to read more about what's the key change that make things work (I am guessing the improved SNR), and whether that key technique can also benefit mutual info estimation.

As someone who has worked extensively in both VI and mutual info estimation, I think this paper worths a highlight for the probabilistic modelling community at ICLR. I think the paper did a reasonably good job beyond just presenting the idea -- their algorithmic choices are also well justified with both mathematical and empirical analyses. The paper also positioned its contribution fairly well by a detailed comparison with many of existing works, except that --

There is another line of work in VI literature that also used the classification loss idea, e.g., see https://arxiv.org/abs/1701.04722 and other related papers published around the same time. It would be useful if the authors further discuss this line of work in camera ready, making their positioning effort complete.

**Additional Comments On Reviewer Discussion:**

Most of the reviewer concerns were addressed in author rebuttal.

In reviewer - AC discussions, reviewers recommended clear accept.

---

> ### Public Comment · ~Daniel_Ward3 · 2025-02-10
>
> Thank you. As suggested, we have included a brief description of adversarial variational Bayes in the camera ready version.

---

### Decision · Program_Chairs · 2025-01-22

Accept (Spotlight)